# Reputations for treatment of outgroup members can prevent the emergence of political segregation in cooperative networks

Brent Simpson [1] ✉, Bradley Montgomery[2] & David Melamed [2,3] ✉

Reputation systems promote cooperation and tie formation in social networks. But how reputations affect cooperation and the evolution of networks is less clear when societies are characterized by fundamental, identity-based, social divisions like those centered on politics in the contemporary U.S. Using a large web-based experiment with participants (N = 1073) embedded in networks where each tie represents the opportunity to play a dyadic iterated prisoners' dilemma, we investigate how cooperation and network segregation varies with whether and how reputation systems track behavior toward members of the opposing political party (outgroup members). As predicted, when participants know others' political affiliation, early cooperation patterns show ingroup favoritism. As a result, networks become segregated based on politics. However, such ingroup favoritism and network-level political segregation is reduced in conditions in which participants know how others behave towards participants from both their own party and participants from the other party. These findings have implications for our understanding of reputation systems in polarized contexts.

Reputation systems are foundational to the high levels of cooperation observed in human populations[1–3], allowing people to select partners with a history of prosocial behavior and avoid those with reputations for selfishness[4,5]. As a result, reputation systems create strong incentives to act prosocially since those with prosocial reputations are more apt to be selected for lucrative cooperative relationships[6,7] and less likely to be excluded from social networks[8]. Thus, in dynamic human networks, where people can sever unwanted ties and form new ones[4,9], reputations lead to clusters of cooperation, thereby increasing the welfare of people who behave more cooperatively[10–12].

But how do reputation systems impact cooperation, clustering, and segregation in human societies characterized by fundamental, identity-based, social divisions? For instance, the contemporary US is characterized by extreme levels of affective political polarization[13,14].

Under such conditions, while it almost certainly pays to have a reputation for cooperating with members of one's own party, it is less clear that it is also beneficial to be known as someone who cooperates with members of the opposing party.

More importantly for current purposes, different types of reputational information—e.g., whether and how reputations track behavior toward outgroup members (e.g., people from an opposing political party) versus only ingroup members (e.g., people from one's own political party)—may not only give rise to different patterns of cooperation with ingroup versus outgroup members. Variations in reputation systems may also influence the formation of ties within versus between social categories. This may lead to network-level segregation based on politics and could help drive political polarization like we see in the contemporary US, where

[1]Department of Sociology, University of South Carolina, Columbia, SC 29208, USA. [2]Department of Sociology, The Ohio State University, Columbus, OH 43210, USA. [3]Translational Data Analytics Institute, The Ohio State University, Columbus, OH 43210, USA. ✉e-mail: bts@sc.edu; melamed.9@osu.edu

Americans report being less likely to have social ties that span across party lines[14].

Here we embed participants in dynamic networks where each tie represents an opportunity to interact in a dyadic prisoners' dilemma situation, i.e., where there is a conflict between individual and collective interest[15,16]. We experimentally vary whether participants have access to others' political identities and the types of reputations people can develop, e.g., whether a person's reputation distinguishes how they treat members of one group versus another. Our first aim was to assess how these manipulations impact both overall cooperation and cooperation with participants from opposing political parties, particularly in the early stages of interaction, i.e., before network dynamics produce high levels of cooperation across conditions. Our second aim was to assess how variation in the types of reputations participants could develop—and the resulting early cooperation patterns—lead to different tendencies to form and break ties to others from one's own versus the opposing political party. We expected these differences would give rise to different levels of segregation based on political identities, rather than cooperative tendencies. Although political identity is our test case, following recent work on politics as a social identity[13,17,18], our findings should contribute to a more general understanding of the effects of reputation systems on cooperation and network-level segregation based on a range of social identities.

Given current levels of distrust and animosity between partisans[14], we wanted to study the effects of different types of reputational information in an environment that is highly conducive to cooperation. Thus, following recent work on dynamic networks[4,5,8], we allow participants to sever existing ties and form ties with new partners over the course of the study. Network dynamics increase cooperation by allowing more cooperative types to sort with one another and avoid being exploited by non-cooperators. As a result, we expected that early differences in cooperation between conditions would ultimately give way to high levels of cooperation across all conditions. But we also expected that early differences in cooperation—both overall cooperation rates and differences in cooperation with ingroup versus outgroup members—would lead to different patterns of tie dissolution and formation across different experimental conditions. This, in turn, would lead to topological differences in social networks between reputation conditions, namely the extent to which the networks showed political segregation.

Understanding the social and informational roots of polarization is important in our current environment, as affective polarization underlies an array of fundamental social problems[19] and poses threats to democracy[20]. Increasing cross-cutting ties is thus critical to democracy itself, at least if accompanied by changing attitudes and institutional reforms[21], as discussed later. By experimentally manipulating whether participants have knowledge of one another's political identities, and the types of reputations people can carry, we aim to shed light on some social conditions that may reduce political polarization while maintaining high levels of cooperation. We focus on four types of reputation systems in networks that vary by experimental condition (see Table 1). In the control condition, participants do not know one another's political identities, but they do have access to others' reputations, namely their level of cooperation with other participants in previous rounds. This condition is in line with most research on reputations in networks[4,5,8]. In the three treatment conditions, participants know one another's political identities. As shown in Table 1, these undifferentiated, parochial, and intra/intergroup reputation conditions differed by whether and how they distinguished a person's treatment of ingroup and outgroup members.

Undifferentiated reputations do not distinguish between interactions with ingroup and outgroup members. Instead, they are determined by a person's level of cooperation in their previous interactions, regardless of who those interactions are with. These types of reputations are powerful drivers of cooperation when social identities

are not known but may hinder cooperation in the presence of strong social cleavages. For instance, we know from extensive prior work that people are more likely to cooperate with ingroup members than outgroup members[22–24]. This tendency may be especially strong when the basis of group identities is political affiliation[25]. But in a polarized environment, reputations that do not differentiate between how a person treats ingroup versus outgroup members may send mixed signals. For instance, a person who is highly cooperative with ingroup members but uncooperative with outgroup members will tend to have a relatively uncooperative reputation with members of their outgroup and ingroup. In environments where most reputations are relatively uncooperative, people will likely default to interacting with ingroup members, given the tendency to trust fellow ingroup members more than outgroup members[23,26,27].

Additionally, undifferentiated reputations do not account for the fact that reputations are often based explicitly on group identities. A long line of theory and research suggests that people are primarily motivated to maintain positive reputations with their ingroup members[28,29]. Further, since humans associate more with fellow ingroup members than outgroup members[30], reputational information in the real world may be more apt to flow between ingroup members. This may create an even stronger motivation to maintain a positive reputation in interactions with ingroup members while reducing the incentive to build cooperative reputations vis-a-vis outgroup members. Indeed, if interactions with outgroup members are comparatively rare, as is increasingly the case with respect to politics in the US, it may be difficult for prosocial reputations toward outgroup members to form at all. We call reputations that are limited to interactions with ingroup members parochial reputations. As detailed more fully below, we expect that parochial reputations will lead to higher levels of cooperation with—and tie formation to—fellow ingroup members, relative to outgroup members.

Unlike undifferentiated reputations (which do not distinguish how people treat ingroup versus outgroup members) and parochial reputations (which only track how people treat fellow ingroup members), people may develop reputations for how they treat ingroup members as well as reputations for how they treat outgroup members. Existing research suggests competing effects of having distinct reputations for ingroup members and outgroup members on intergroup cooperation and the segregation of networks based on group identities.

One line of research suggests that reputation systems that differentiate the treatment of ingroup and outgroup members will reduce intergroup cooperation and increase network-level segregation. In particular, social identity theory[26] assumes that people seek to maximize the relative advantage of their ingroup over outgroups. From this perspective, while positive treatment of ingroup members is important, people may also prefer to interact (and cooperate) with ingroup members who do not cooperate with outgroup members since non-cooperation with outgroup members can help maximize the ingroup's relative advantage[31]. The ability to track how people treat outgroup members may give additional force to any such tendency, since people may shun or avoid ingroup members who have reputations for cooperating with outgroup members. Indeed, ingroup members who retaliate against outgroup members in intergroup conflict accrue status rewards within groups[32]. Motivations to exploit—or at least avoid—cooperation with outgroup members may be especially strong when group membership is counter-normative[33] or morality-based, as is the case for political parties[34]. This suggests we may observe particularly high levels of homophily-based cooperation and political segregation in networks when reputation systems differentiate between the treatment of ingroup and outgroup members.

But another line of research on unbounded indirect reciprocity[35,36] suggests that distinct reputations for ingroup and outgroup members will promote cooperation with both ingroup and outgroup members

**Table 1 | Summary of experimental conditions and theoretical expectations**

| Condition | Political identity | Reputations | Theoretical expectations |
|---|---|---|---|
| Control | Hidden | Based on avg. number of monetary units participants gave to alters in previous interactions | Provides baseline. Sorting will be based on cooperation |
| Undifferentiated reputations | Visible | Based on avg. number of monetary units participants gave to alters in previous interactions. Rep. does not distinguish between interactions with ingroup- and outgroup members. | Early cooperation rates will show ingroup favoritism and the network will become segregated based on politics. |
| Parochial reputations | Visible | Based on avg. number of monetary units participant gave to their own ingroup members in previous interactions. | Early cooperation rates will show ingroup favoritism and the network will become segregated based on politics. |
| Intra/Intergroup reputations | Visible | Two reputation scores, one based on avg. number of monetary units participant gave to their ingroup and another based on avg. number of monetary units participant gave to their outgroup. | Based on social identity theory, early cooperation rates and network segregation will be similar to those found in undifferentiated and parochial reputation conditions. Based on unbounded indirect reciprocity, early cooperation rates will show less ingroup favoritism and networks will show less political segregation compared to the undifferentiated and parochial reputation conditions. |

**Table 2 | Prisoners' dilemma incentive structure**

| Player 1's choice | Player 2's choice | |
|---|---|---|
| | Cooperate (Send 50, which is doubled) | Defect (Send 0) |
| Cooperate (Send 50, which is doubled) | Player 1 earns 50 Player 2 earns 50 | Player 1 loses 50 Player 2 receives 100 |
| Defect (Send 0) | Player 1 earns 100 Player 2 loses 50 | Player 1 earns 0 Player 2 earns 0 |

For simplicity, this table gives the payoffs for maximal cooperation (sending all 50 monetary units to that alter for that round) and maximal defection (sending nothing to that alter for that round). Note that this payoff structure satisfies both inequalities required for a Prisoners' Dilemma. First, $T$ (ego's payoff when ego defects and alter cooperates) = 100 > $R$ (ego's payoff when both cooperate) = 50 > $P$ (ego's payoff when both defect) = 0 > $S$ (ego's payoff when ego cooperates and alter defects) = –50. Second, $2R$ (100) > $T + S$ (50).

and produce greater integration across party lines. Several studies, including an investigation across 17 societies[35], show that people are more cooperative with both ingroup members and outgroup members when their reputations are at stake[36]. From the unbounded indirect reciprocity perspective, while we should expect that people will value interactions with ingroup members, the ability to form positive reputations with outgroup members will also lead to higher levels of cooperation and tie formation across group boundaries. Thus, intra/intergroup reputations that differentiate the treatment of ingroup and outgroup members may promote cross-party cooperation over and above either parochial reputations (which limit the incentive to cooperate with outgroup members) or undifferentiated reputations (which limit the ability to signal that one is cooperative with outgroup members). Table 1 summarizes these theoretical arguments.

To test these expectations, we embedded participants in networks where ties represented the opportunity to cooperate for mutual gain, but also created opportunities to exploit alters' cooperation[5]. We manipulated political identities' visibility and the type of reputation systems that were present. Following prior work on reputations in networks[5,8,10,37], reputation scores were based on objective information, namely the average number of monetary units a participant donated to their alters in the previous three rounds. That is, rather than a subjective evaluation of another person, in our design reputations were objective indicators of the extent to which the person was cooperative (and, depending on condition, with whom). This type of reputation is sometimes referred to as an image score[38–41]. Thus, following prior work on cooperation in networks and for simplicity, the type of reputation we employed did not distinguish between whether a person cooperated (or did not cooperate) with a cooperative versus uncooperative alter, a type of reputation system called standing[42,43]. Our design therefore rules out higher-level reputational norms (e.g. "cooperate only with cooperative others"). Thus, in the control condition, where participants did not have information on one another's political party, a person's reputation was determined solely by their level of cooperation with their alters, as in previous work[4,5,8]. Political identities were visible in the other three conditions, and reputations corresponded to the undifferentiated, parochial, and intra/intergroup reputation systems, as described above (see also Table 1).

Here we show that when participants know each other's political identities, early cooperation patterns show ingroup favoritism. As a result, networks become segregated based on participants' self-reported political affiliations. However, both ingroup favoritism and network-level political segregation are reduced when participants carry distinct reputations for the treatment of ingroup and outgroup members, i.e., in the intra/intergroup reputations condition. These findings yield insights into the conditions under which political identities can hamper the capacity for reputation systems to promote cooperation and ties between groups, and what kinds of reputation systems may prove robust to such detrimental ingroup/outgroup effects.

## Results
### Cooperation rates
Each participant started the study with an endowment of 1000 monetary units. Following previous work, each network tie represented an opportunity to interact in an iterated prisoner's dilemma[4,5,37]. Specifically, participants decided how many of the 50 monetary units to send to each of their alters, in increments of 10, which were subtracted from the sender's pool of monetary units. Thus, 0 represented complete defection and 50 represented maximal cooperation. Any amount sent was doubled and awarded to the partner. For example, if a sender chose to give 50 monetary units to their partner in a round, the sender would lose 50 monetary units from their endowment and the receiver would receive 100 monetary units from that relationship in that round. Participants made an independent decision for each alter to whom they were tied[8,37]. Table 2 shows that each decision conforms to the payoff structure of the prisoners' dilemma[15].

Unsurprisingly, given the powerful impact of dynamic networks on cooperation, Fig. 1A shows that by the end of the study, cooperation rates were high in all conditions, and differences in cooperation with ingroup versus outgroup members had largely disappeared. But we argued that early cooperation patterns, specifically differences in cooperation with ingroup and outgroup members, and corresponding ingroup favoritism in partner selection, would shape eventual network-level outcomes. To this end, we modeled cooperation

between participants in the first eight rounds, i.e., before equilibrium. As illustrated in Fig. S2 we find that early overall cooperation rates were highest in the control and inter/intragroup conditions, and lowest in the parochial condition, with intermediate levels in the undifferentiated condition.

In addition to overall cooperation rates, ingroup favoritism in cooperation in the first eight rounds varied by condition ($\chi^2 = 188.52$, DF = 3, $p < .001$, comparing Models 1 and 2 in Table 3). Figure 1B displays marginal means from Model 3 in Table 3, illustrating differences in cooperation with ingroup versus outgroup

members after controlling for direct reciprocity[44]. We find an inverse relationship between the amount given on average and the size of the ingroup favoritism effect. In the parochial reputation condition, where overall giving is lowest, we see the largest ingroup favoritism effect. Similarly, in the undifferentiated condition, with the second-lowest overall giving, we see the second-largest ingroup favoritism effect. We expected these early differences between conditions would shape network dynamics and the corresponding network topology, ultimately resulting in different patterns of network-level segregation.

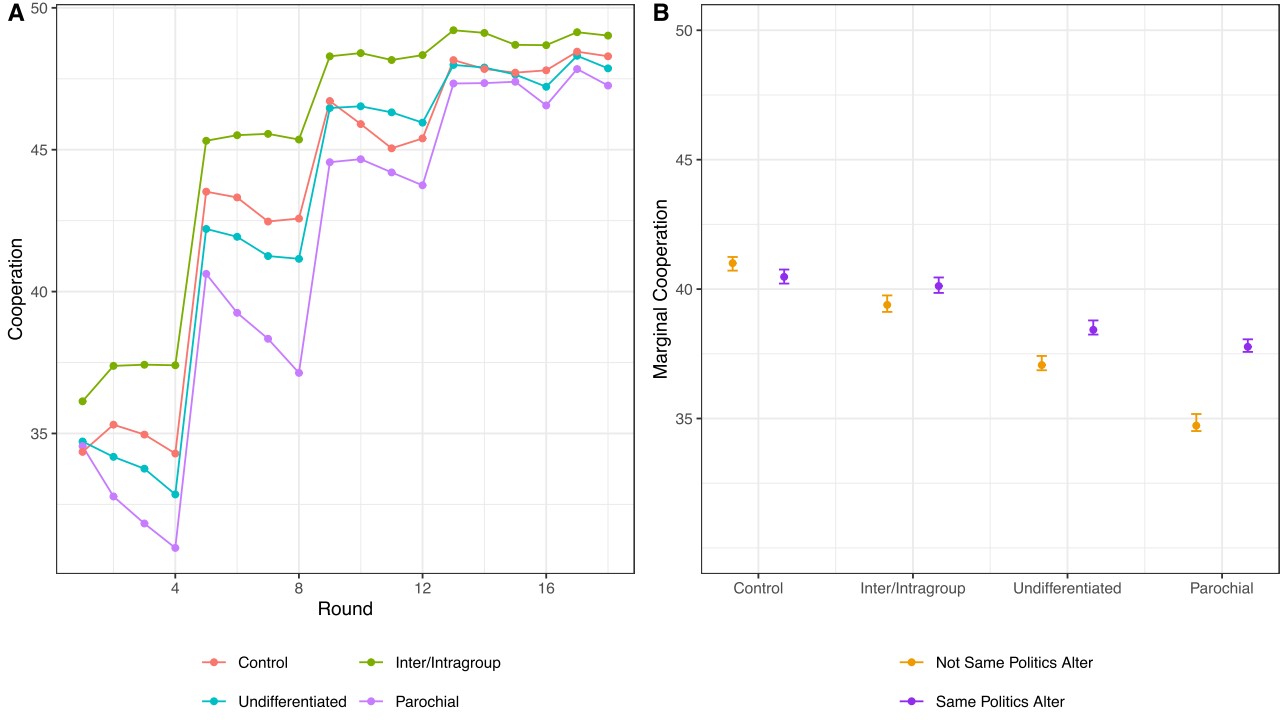

**Fig. 1 | Average cooperation.** Cooperation is measured by number of monetary units sent in each round. All rounds for each experimental condition are given in (**A**). The sample size of **A** is 67,774 instances of cooperation, nested in rounds, nested in participants. Marginal cooperation (number of monetary units sent) for rounds 1–8 (**B**) by experimental conditions and whether the political identity of alter is the same as alter. Margins in **B** drawn from Model 3 in Table 3. The sample size for Model 3 in Table 3 is 26,053 participant-round-alters. Error bars are 95% confidence intervals computed via bootstrapping.

**Table 3 | Summary of linear mixed models predicting how much the participant cooperated in rounds 1–8**

| | Model 1 | Model 2 | Model 3 |
|---|---|---|---|
| Parochial reputations[a] (*P*) | −5.771 (0.999) | −9.389 (<0.001) | −5.764 (0.999) |
| Inter/Intragroup reputations[a] (*I*) | −1.712 (<0.001) | −3.076 (<0.001) | −1.442 (0.251) |
| Undifferentiated reputations[a] (*C*) | −4.012 (0.999) | −6.093 (<0.001) | −3.592 (0.999) |
| Homophily (*H*) | 2.482 (<0.001) | −0.735 (0.072) | −0.456 (0.161) |
| *P* × *H* | | 5.980 (<0.001) | 3.171 (<0.001) |
| *I* × *H* | | 2.112 (<0.001) | 1.088 (0.063) |
| *C* × *H* | | 3.351 (<0.001) | 1.643 (0.009) |
| Intercept | 39.801 (0.999) | 41.807 (<0.001) | 20.243 (0.999) |
| Direct reciprocity | | | 0.534 (<0.001) |
| Variance components | | | |
| Decision | 6.495 | 6.493 | 5.797 |
| Participant | 10.837 | 10.834 | 4.610 |
| Network | 3.762 | 3.803 | 2.057 |
| AR(1) | 0.038 | 0.038 | 0.038 |

Inference is based on 1000 permutations of the outcome within network-rounds.
[a]Reference category is the No Politics/Control condition. *N* for Models 1 and 2 is 31,442, and for Model 3 it is 916 26,053 (round 1 values are missing for Direct Reciprocity). All tests are two-tailed.

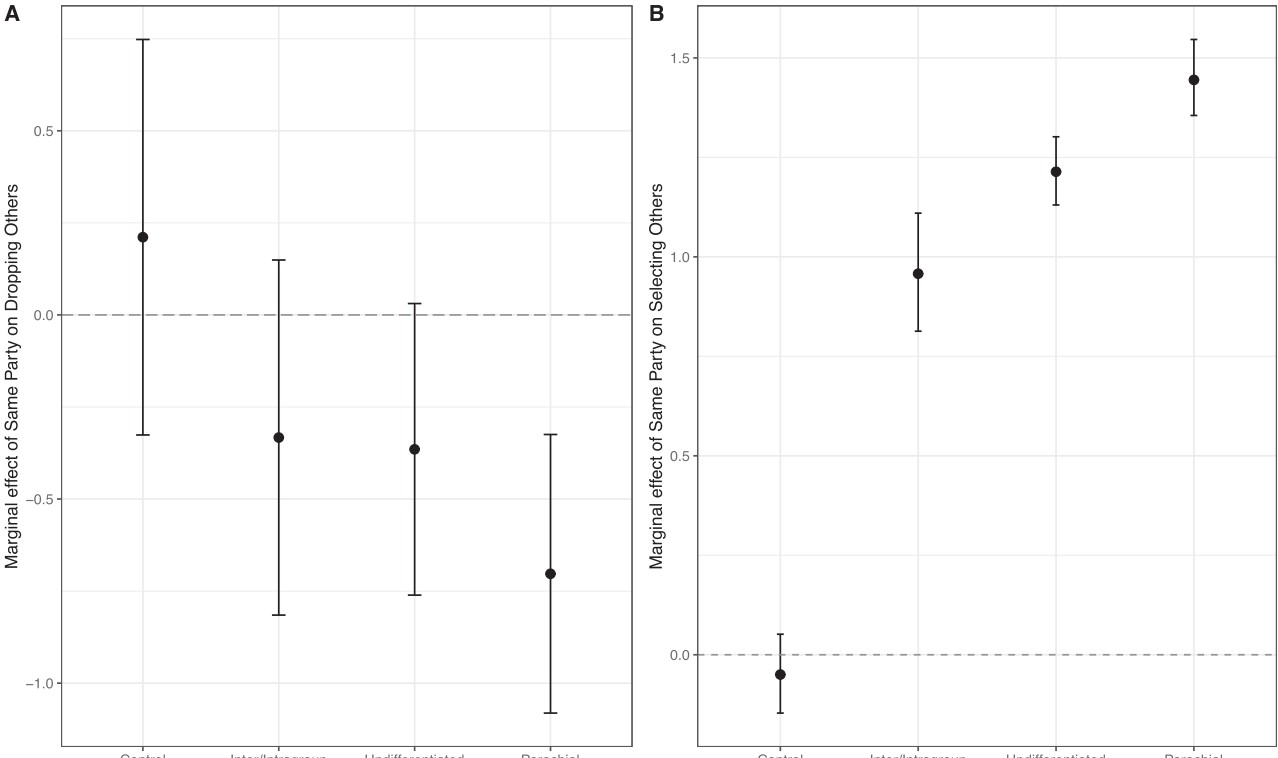

**Fig. 2 | Effects of political homophily on severing and proposing ties. A** Marginal effect of political homophily on selecting which alter to drop. Margins drawn from Table S2, Model 2. The sample size for Model 2 in Table S2 is 1251 alters dropped out of 5928 alternatives. Error bars are 95% confidence intervals computed via the Delta method. **B** Marginal effect of political homophily on which alter to select. Margins drawn from Table S3, Models 1–4. The sample size for Model 1 is 268 selections from 4612 alternatives. The sample size for Model 2 is 380 selections from 7711 alternatives. The sample size for Model 3 is 412 selections from 7711 alternatives. The sample size for Model 4 is 208 selections from 3062 alternatives. Error bars are 95% confidence intervals computed via bootstrapping.

## Severing ties

Participants had the option to sever one tie and propose a new tie to another person in the network every four rounds, for a total of four opportunities to cut and add ties. Participants were not informed how often or when tie updates would occur. In tie update phases, any tie could be severed unilaterally but proposed new ties required approval by the selected other. This is consistent with closely related work[8,37,45] as well as sociological conceptions of tie formation[46], which assume that it takes two people to form a relationship, but one to end it.

Given the variation in cooperation patterns between conditions (Figs. S2 and 1B), we should expect to find condition-level differences in participants' tendencies to sever ties. Figure S4 shows differences in probabilities of dropping any alter by experimental condition (see also, Fig. S3. Table S1 and corresponding SI discussion). These results show that participants in the intra/intergroup reputations condition were consistently less likely to drop alters than participants in all other conditions, including the control. This creates less opportunity for topological change in the intra/intergroup reputations networks, compared to networks in the other conditions.

Conditional on dropping an alter, participants then selected which alter to drop. Across all conditions, participants tended to retain more cooperative alters (Fig. S5A; Table S2, Model 1, $\beta = -0.048$, $p < .001$, 95% confidence interval $= -0.054$, $-0.042$, z-test). But how did politics impact who was dropped in the three politics treatment conditions? Figure 2A illustrates the effect of sharing a political identity on selecting which alter to drop. Participants in the parochial reputations condition were significantly less likely to drop ingroup members relative to the control ($\beta = -0.914$, $p = 0.002$, 95% confidence interval $= -1.469$, $-0.359$, z-test). Participants in the other two experimental conditions were not statistically different from participants in the control condition (undifferentiated reputations $\beta = -0.58$, $p = 0.053$, 95% confidence

interval $= -1.143$, $0.017$, z-test; intra/intergroup reputations $\beta = -0.54$, $p = 0.09$, 95% confidence interval $= -1.161$, $0.081$, z-test).

To put these effects in context, consider results for the parochial reputations condition, which showed the strongest tendency for participants to sever ties to outgroup versus ingroup members (Fig. 2A). To offset this homophily effect, an outgroup alter would have to give ego 14.4 more monetary units than an ingroup alter to have an equal probability of being dropped. More generally, Fig. S6 shows how many monetary units participants in the parochial reputations condition would have needed to receive from ingroup and outgroup members for the two to stand equal chances of being dropped. These results shed light on the relative "value" of ingroup over outgroup ties, particularly in the parochial reputation condition.

Summing up, participants in the intra/intergroup reputations condition were the least likely of all participants to drop alters, reflecting the high cooperation rates in this condition in early rounds. When participants in this condition did sever ties, tie-deletion was based on levels of cooperation. In contrast, paralleling differences in early cooperation rates toward ingroup and outgroup members, participants in the undifferentiated reputations and parochial reputations conditions were substantially more likely to drop alters, and those in the parochial reputations condition were more likely to drop outgroup members. We show below that these dynamics have important consequences for the level of network segregation that occurred in the intra/intergroup reputation conditions versus the other two conditions where politics was visible.

## Tie formation

After severing a tie, participants then proposed a new tie to a prospective alter. Following prior work[5,8], any proposed tie had to be

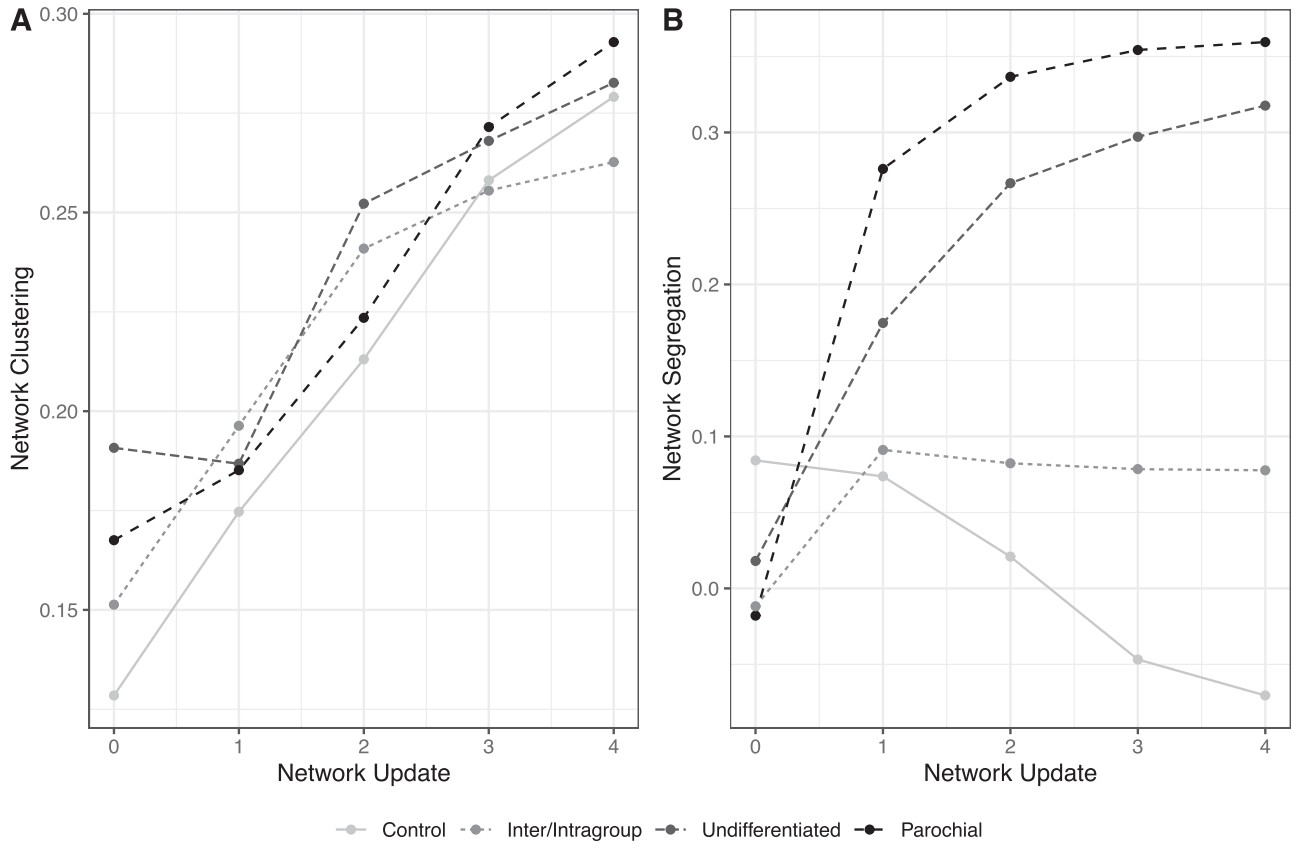

**Fig. 3 | Changes in networks over time.** Network clustering (**A**) and segregation (**B**) at initiation and following each network update by experimental condition. Networks in all conditions became more clustered over time (**A**). But whether networks became more segregated varied by condition (**B**).

confirmed by the prospective alter. As illustrated in Fig. S5 and Table S3, participants tended to propose ties to cooperative others, and thus those who had prosocial reputations. More importantly for our purposes, across all three conditions where politics were visible, we found that people were more likely to propose ties to ingroup members than outgroup members (Fig. 2B). The effect was strongest in the parochial reputations condition (Table S3, Model 3, $\beta = 1.445$, $p < 0.001$, permutation-based inference), second strongest in the undifferentiated reputations condition (Table S3, Model 2, $\beta = 1.214$, $p < 0.001$, permutation-based inference), and weakest, but still positive and significant, in the intra/intergroup reputations condition (Table S3, Model 4, $\beta = 0.958$, $p < 0.001$, permutation-based inference). To offset the homophily effect in the parochial reputation condition, for example, an outgroup member would have to have a 23-point higher ingroup reputation than an otherwise comparable ingroup member. An outgroup member in the undifferentiated reputation condition would need to have an 18.4-point higher reputation than an otherwise comparable ingroup member to be selected. Figure S7 shows this result

across all values of reputation scores by ingroup and outgroup members for the undifferentiated reputation condition.

Despite these differences between conditions in the tendency to propose ties to ingroup versus outgroup members, we do not find significant differences in conditions in the tendency to accept tie proposals from ingroup versus outgroup members ($\beta = 0.075$, $p = 0.459$, permutation-based inference). Instead, tendencies to accept a tie proposal were primarily driven by the proposer's reputation (Table S4, $\beta = 0.084$, $p < 0.001$, permutation-based inference).

### Network clustering and segregation

Here we examine the clustering and segregation of networks over time. Clustering is defined as the probability that adjacent nodes are connected[47], and segregation is defined as having fewer between-group ties than expected by chance given the density of the network[48]. Networks in all four conditions linearly increased in clustering over the course of the study (Fig. 3A). And as shown in Model 1 of Table 4, there were no statistically significant differences in network clustering between conditions at the end of the study. Importantly, however, the basis of clustering depended on experimental conditions. As already demonstrated, there was less network change in the intra/intergroup reputation condition (Fig. S4). Further, when network change did occur, political homophily drove tie deletion and formation differentially across experimental conditions (Fig. 2). Finally, while participants preferentially proposed ties to fellow ingroup members in all three politics conditions, this tendency was weakest in the intra/intergroup condition.

As a result of different patterns in tie deletion and tie formation across conditions, a significantly higher level of political segregation emerged in the parochial and undifferentiated reputations conditions than in the intra/intergroup reputations and control conditions

**Table 4 | Summary of OLS regression models predicting network clustering and network segregation**

| | Clustering | Segregation |
|---|---|---|
| Parochial reputations[a] | 0.014 (0.746) | 0.430 (<0.001) |
| Inter/Intragroup reputations[a] | −0.016 (0.699) | 0.148 (0.092) |
| Undifferentiated reputations[a] | 0.004 (0.934) | 0.388 (<0.001) |
| Constant | 0.279 (<0.001) | −0.070 (0.252) |

Each network represents a single case. For consistency with other analyses, parametric p-values are reported in parentheses.
[a]Reference category is the control condition. N is 40 networks. All tests are two-tailed.

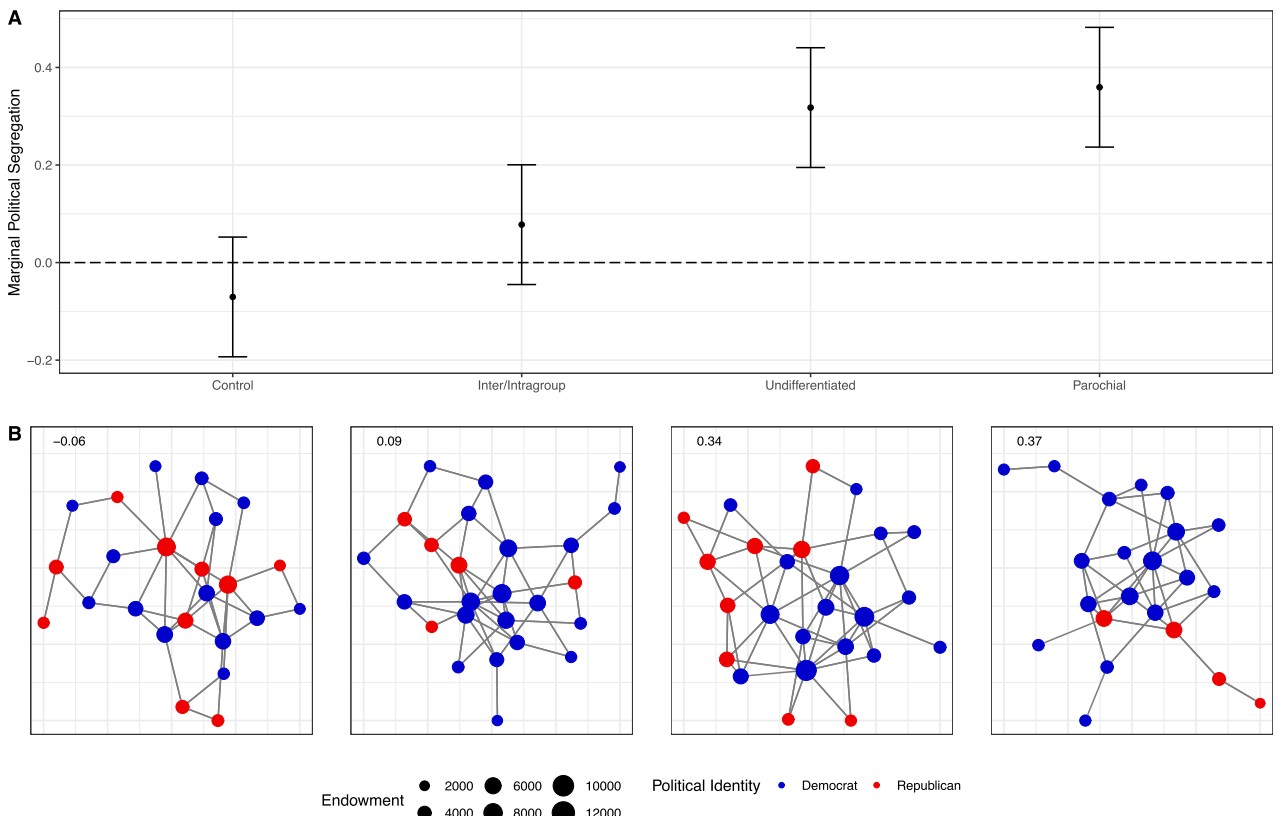

**Fig. 4 | How political segregation differs by condition.** Marginal segregation between Democrats and Republicans at the conclusion of the study by experimental condition (**A**). Margins drawn from Table 4 (*N* = 40 networks). Error bars are 95% confidence intervals computed via the Delta method. To illustrate, **B** shows networks in each condition with observed network segregation close to the estimated marginal mean from (**A**). The networks in **B** were selected for having segregation patterns typical of the estimated means from (**A**). They are not representative of other patterns within conditions. For instance, while the Inter/Intragroup network in **B** has fewer Republicans than the Control and Undifferentiated networks, this is not true of this condition more generally. Indeed, Fig. S25 shows that Republicans are less likely to become isolated from the networks in the Inter/Intragroup condition than in the two other conditions. See Supplementary Note 2 and Figs. S28–S31 of the SI for visualizations of all 40 networks in our experiment, broken down by condition.

(Fig. 3B). This is also shown in Model 2 of Table 4, which presents a summary of an OLS regression model predicting network-level segregation. More specifically, as illustrated in Fig. 4A, simply making political affiliations visible caused a significant increase in political segregation by the end of our study, but only in the undifferentiated and parochial reputation conditions. That is, we did not find any statistically significant tendency for networks where people have distinct reputations for their treatment of ingroup and outgroup members to become more politically segregated over time.

To illustrate the magnitude of our experimental manipulations on political segregation, Fig. 4B shows one network from each of our conditions at the end of the study with approximately the same political segregation as the margins in Fig. 4A. Note that self-identified Republicans are more integrated in the networks in the control and intra/intergroup conditions (the two on the left) compared to the parochial and undifferentiated conditions (the two on the right). Supplementary Note 2 of the SI contains visualizations of all 40 networks, broken down by experimental conditions.

## Discussion

One of the most powerful solutions to the array of cooperation problems humans face is our ability to track one another's reputations[1–3,49]. Reputation systems alter network dynamics by increasing the tendency for people who act more prosocially to attract partners and form productive clusters of cooperation. But it is not clear how reputations work when social environments are characterized by fundamental, identity-based cleavages. A key example is the political sectarianism in the contemporary US, marked by distrust, outgroup hate, and decreasing ties across party lines[19]. Here we hypothesized that whether and how such identity-based polarization develops or persists depends on the types of reputations that people carry.

We studied three different types of reputation systems (undifferentiated, parochial, and intra/intergroup) that differ in whether and how a participant's reputation tracked their interactions with other participants who identified with the same versus different political parties. In parochial reputation systems, participants' reputations were determined solely by how they treated their fellow ingroup members in prior rounds. We expected that parochial reputation systems would lead to strong homophily effects, both in terms of early levels of cooperation and in network dynamics. One might expect that cooperation and tie formation would be less subject to ingroup favoritism if people had politically undifferentiated reputations that did not distinguish between the treatment of their fellow partisans versus outgroup members. But we argued that undifferentiated reputations send ambiguous signals and may therefore lead people to default on category membership as a basis for trust, cooperation, and partner selection. Our findings are consistent with these expectations.

We built on several streams of theory and research to outline competing predictions for our intra/intergroup reputation conditions, where reputations distinguished between the treatment of one's ingroup and outgroup members. One line of research suggests that such reputation systems will lead to less cooperation with outgroup members and will increase network-level segregation. A key rationale

for this line of research, as noted earlier, is social identity theory[26,31], which assumes that people will seek to maximize relative advantage over outgroups. From this perspective, we should have observed high levels of ingroup favoritism in cooperation early in the study, as well as network-level segregation based on politics over time. But an unbounded indirect reciprocity approach, supported by our findings, suggests instead that distinct reputations for the treatment of ingroup and outgroup members will promote cooperation with both ingroup and outgroup members early on and lead to higher levels of integration across party lines[35,36]. Based on this perspective, we argued that reputations that differentiate the treatment of ingroup and outgroup members would promote cross-party cooperation over and above either parochial reputations (which limit incentives to cooperate with outgroup members) or undifferentiated reputations (which limit the ability to signal that one is also cooperative with outgroup members).

Although we found significantly higher levels of political segregation in the parochial and undifferentiated reputations conditions, the level of political segregation in the intra/intergroup reputations condition was not statistically different from networks in the control condition, where political identities were unknown. That is, rather than sorting on self-identification as a Democrat or Republican, as happened for the parochial and undifferentiated reputation conditions, participants in the intra/intergroup reputations condition-like control participants tended to sort on cooperation.

These network-level segregation effects followed differences between conditions in early cooperation patterns and network dynamics. In particular, participants in the intra/intergroup reputations condition not only showed higher overall levels of cooperation in early rounds; unlike the other two politics conditions, ingroup favoritism in this condition did not statistically differ from the level of ingroup favoritism in the control condition. Indeed, as shown in Fig. 1A, overall cooperation in the intra/intergroup reputations condition started out higher than in all other conditions and remained so throughout the duration of the study. That this difference emerged from round 1 suggests that it is not something that participants learned over time through trial and error. Rather, given the information content of reputations in this condition, participants likely considered it advantageous to cooperate with outgroup members, rather than solely with fellow ingroup members. This provides some potential insights into the design of reputation systems, an issue we take up below.

As a result of differences in cooperation in early rounds, participants in the intra/intergroup reputations condition were less likely than those in other conditions to sever ties; when they did sever ties, they were not more likely to cut ties to outgroup members. Finally, although participants in all three politics conditions tended to preferentially propose new ties to ingroup members, this tendency was weakest in the intra/intergroup reputations condition.

These results are surprising from the standpoint of social identity approaches. After all, having information on how people treat the outgroup might be expected to create an incentive to exploit outgroup members, as doing so would be valued by fellow ingroup members in an acrimonious political environment. Indeed, prior work suggests that such effects are most likely when group identities are morality-based, as is the case for political parties[34]. This suggests that participants should have been especially motivated to exploit outgroup members, and perhaps even shun ingroup members who developed reputations for cooperating with outgroup members. But this did not happen in our experiment. In fact, we observed preferential attachment to fellow ingroup members with reputations for being prosocial to outgroup members (Table S3).

A key question for future research is whether these findings can be extended to reduce prejudice and distrust across party lines. Intergroup contact theory[50,51] suggests that interactions across group boundaries—especially those centered on common goals, as in the cooperative interactions we studied—can reduce prejudice and distrust of outgroup members. Furthermore, these effects extend beyond the specific outgroup members involved in the interactions to the outgroup as a whole[51]. A simple but important extension of this project would involve replicating our design but measuring general attitudes towards the opposing political party at the conclusion of interactions. Based on higher levels of political integration we observed in the intra/intergroup networks, we would expect more favorable attitudes across party lines in this condition.

Another key issue for future research is an exploration of the domains to which these findings apply. In contrast to many social media platforms, which often create stronger incentives for highly partisan behavior and derogation of outgroup members, participants enrolled in our studies for monetary gain. Similarly, the stakes of interactions in our study were monetary rather than political outcomes. In highly polarized environments, interactions over politically relevant outcomes are more apt to be viewed as zero-sum[52] such that any gain for "their" side is necessarily a loss for "our" side. But in our study, material rewards for mutual cooperation with the political outgroup were clearer, making cooperation more likely.

To be clear, political identities also affect decision-making when the stakes are monetary. For example, empirical evidence shows that partisanship affects the wages demanded by workers, as well as consumers' willingness to purchase items[53]. Our findings complement this prior work on how partisanship infiltrates cold economic calculations by showing how cooperation patterns can lead to the emergence of political segregation in networks, and how the extent of that segregation depends on the types of reputations that can be formed.

It remains to be seen whether an alternate version of our experiment where rewards are more symbolic, as is more typical of social media, or where cooperation is beset by zero-sum thinking, would generate different results. It may be that we would observe an overall increase in political segregation across all political identity conditions, but the differences between reputation conditions would remain. It is also possible that the differences we observe between conditions would not withstand the more acrimonious partisanship and zero-sum thinking common among partisans on social media. Indeed, a recent review[21] of interventions to reduce partisan animosity concludes that any single intervention is unlikely to be effective by itself, given that partisan animosity arises from, and is reinforced by, processes at three different levels—thoughts (e.g., misperceptions of the other side), relationships (relative absence of ties across party lines) and institutions (norms and political structures that promote partisanship). Thus, beyond the question of whether our findings would apply in more acrimonious, zero-sum, contexts, our work only addresses how different types of reputation systems impact relationships across party lines. It does not address thoughts or institutions. And while a central tenet of intergroup contact theory is that ties to outgroups can reduce outgroup prejudice and misperceptions (i.e., change thoughts), the arguably more difficult problem of changing institutions would remain.

Even if our findings do not directly generalize the hyper-polarized case to contemporary American politics, political identities are instances of social identities[13,17,18]. As such, our findings address a key question at the intersection of two largely distinct literatures: on the one hand, reputations have powerful effects on prosocial behaviors[38,54], but it is not clear whether these effects are robust to the well-known tendency for people to cooperate[55] and preferentially sort with[56] fellow ingroup members. As a result, we do not know from prior work whether and when networks segregate based on cooperative dispositions versus shared identities. Our results suggest that social identities can shift the tendency for networks to sort based on cooperative dispositions to sorting based on identities, but the extent to which this result holds depends on what types of information reputations carry. As a result, these findings provide insight into the effective design of reputation systems when populations risk segregating into clusters

based on key group identities. As discussed earlier, people may have greater access to reputational information about ingroup members. Particularly given ingroup favoritism tendencies, this risks high levels of segregation based on identity, as shown in our parochial reputation condition. One might expect that undifferentiated reputation systems, where individuals develop reputations that do not distinguish between the treatment of ingroup and outgroup members, would circumvent identity-based segregation. But our findings suggest otherwise. Like the parochial reputation condition, our undifferentiated reputation networks tended to sort based on identities, resulting in clusters of self-identified Democrats and Republicans.

But again, it remains to be seen the extent to which our findings are scope limited to settings that are relatively conductive to cooperation with outgroup members. Similarly, while past research shows that behavior in the prisoner's dilemma and related behavioral games predict behaviors in more natural situations that pose conflicts between individual and collective interests[57–59], an important question for follow-up work is whether the patterns we observed in our stylized experiment translate to trust and cooperation in real-world networks.

Finally, it is important to address how our findings hold up when reputations form via the subjective assessments of network members. That is, following prior work on reputations in networks[4,5,8,37], reputations in our study were based on objective behavior. But if people evaluate the same behavior by ingroup and outgroup members differently, this may have implications for ingroup vs. outgroup reputations and thus limit the scope of our findings. Importantly, however, research using behavioral games to study politics[60] shows that behavior (e.g., extending trust) provides individuating information that reduces ingroup/outgroup differences, suggesting that people will interpret a highly cooperative behavior from an outgroup member as favorably as they would a highly cooperative behavior from an ingroup member. If so, more subjective bases of reputations would likely yield results like those we observed here. Whether this is so could be addressed in future research.

That specific behaviors provide individuating information that reduces ingroup favoritism may also help explain the finding that participants extended tie requests to ingroup members more than outgroup members, but no such favoritism emerged for the acceptance of these tie requests. It may be that recipients of tie requests from outgroup members viewed these requests as a sign of trust or goodwill, thus minimizing the favoritism we observed for tie requests. Given that such a process would constrain the level of political segregation we observe in networks, future research should test whether individuating information explains this pattern of results.

Summing up, here we drew on social psychological approaches to fundamental identities, research on reputations, and insights into social networks to ask how different types of reputation systems alter the formation of cooperative ties across group boundaries and the corresponding emergence of identity-based segregation in networks. The results of our experiment, which addressed political identities in particular, suggest that a tendency for reputations to form around the treatment of ingroup members only or an undifferentiated reputation system that ignores group boundaries can both lead to politically segregated networks. In contrast, a reputation system that allows people to develop distinct reputations vis-a-vis ingroup and outgroup members promotes levels of cross-party ties similar to the control condition where political identities are unknown. These findings help clarify how reputations work in environments where important social identities are known, how dynamic networks may segregate on identities rather than cooperative tendencies, and the conditions that favor more ties across group boundaries.

## Methods
The methods are built on the broader literature on cooperation in dynamic networks[5,8,61–66]. A total of 1073 participants from the United

States were recruited from Prolific (prolific.co) during the spring and summer of 2022, and embedded in 40 dynamic networks (average initial network size = 26.8). Participants self-selected for the study by following a link from Prolific to a custom application for data collection. Specifically, following closely related work[8,65], we used a version of Breadboard[67] that was tailored to our experiment. All procedures were approved by the Institutional Review Board at the University of South Carolina.

Because Democrats are overrepresented on platforms like Prolific[68,69], we needed to ensure that our networks were not populated solely with Democrats. While Prolific does not allow screening on political parties, it does allow screening on whether participants identify themselves as liberal or conservative. In the past, self-identified liberals and conservatives were distributed relatively equally between the Democratic and Republican parties. But contemporary liberals and conservatives are very likely to identify as Democrat and Republican, respectively[19]. Thus, for each experimental session, we created two Prolific studies: one for liberals and one for conservatives and moderates. (No mention was made of politics in the links to the studies.) Both Prolific studies provided the same description of the experiment and provided the same study link. In the liberal version of the study, we allowed a maximum of 15 people to follow the link from Prolific to our experiment. In the conservative and moderate version of the study, we allowed 25 people to follow the link. The study needed at least 20 participants for the interaction phase to begin. Thus, allowing a maximum of 15 liberals (as self-identified in their Prolific profiles) to enter the session guaranteed we would never have a session with Democrats only.

Participants first completed an informed consent form. Thereafter, participants in the three experimental conditions indicated their political orientation on a six-point scale, with 1 being "much closer to the Republican Party" and 6 being "much closer to the Democratic Party." We classified participants as Republican if they responded 1–3 and Democrat if they responded 4–6. We employed a six-point scale to avoid having many participants classify themselves as Independents. It is possible that forcing participants who consider themselves more independent to classify themselves as either a Democrat or Republican crates a more conservative test of the arguments outlined above. In the three conditions where political identities were visible, each participant's network node was color-coded to match their political orientation, red for Republicans and blue for Democrats. Those in the control condition were asked about their political orientation in the post-study questionnaire. Thus, control participants' nodes did not denote their political orientation.

Each participant then read detailed instructions on decision-making and how network dynamics would occur[70]. Following the experimental instructions, participants were asked five questions to assess comprehension. Participants needed to correctly answer at least four of these questions to move on to the actual experiment. At least 20 participants who successfully passed the comprehension check were required for the session to proceed. Between 20 and 28 participants were admitted into the study in each session. If fewer than 20 participants passed the comprehension check, the session was canceled. The Supplementary Methods section of the SI includes screenshots from the experimental instructions, comprehension check questions, and the actual experiment.

Prior work shows that initial network topology shapes cooperation patterns[71]. Thus, following closely related work[4,8,37,45,56,65,66], initial networks were random (Erdös–Rényi) graphs with a density of 0.167, or about 4.5 ties per node on average. During the cooperation phases, participants only saw the alters to whom they were connected, rather than the entire network of 20–28 nodes.

Each participant started the study with 1000 monetary units (1000 monetary units = $1). Participants' total monetary units and their alters' total monetary units were visible throughout the study. Previous

work on cooperation in networks differs in whether participants make a binary decision to cooperate or defect[4,5,9], or whether cooperation is measured continuously, allowing degrees of cooperation or non-cooperation[45,66,72]. Because continuous measures allow more statistical power[73] and because cooperation in the real world is often a matter of degree, we measured cooperation on a continuous scale. Prior work also differs in whether each participant makes a single decision that affects all alters[4,5], or makes separate decisions for each alter[8,9]. Given our focus on differences in the treatment of ingroup vs. outgroup members, our participants made separate decisions about the number of monetary units to give to each of their alters. That is, participants did not have to send the same number of monetary units to each alter. After making independent decisions about how much to cooperate with each of their partners, each participant was told how much they received from each of their partners that round. The monetary units sent were deducted from the participant's total number of monetary units, and any monetary units received from alters (i.e., after being doubled) were added to the participant's total number of monetary units.

Participants had the option to sever one tie and propose a new one every four rounds. Prior work shows that when there are too many or too few opportunities to alter ties, it limits the ability of cooperators to isolate themselves from defectors. Allowing participants to sever one tie every three rounds is sufficient to alleviate this concern[5,66]. Closely related studies tend to follow this design[37,45]. Our experiment followed these procedures but had tie updates every fourth round, rather than every third, to prevent the experiment from ending on a tie update round with no option to make decisions vis-à-vis new partners immediately after the tie update.

During the tie update phases, participants first decided whether to sever one of their ties. If a participant chose not to sever a tie, they were not given an opportunity to request a tie with another player during that tie update phase[8,37,45]. If a participant chose to sever a tie, they were shown their current alters' total number of monetary units and the number of monetary units each alter donated to the participant in the previous round. (Following previous work[8,37,66], participants were not given information on what their alters gave to others during the decision-making rounds. Doing so would have likely resulted in information overload.) Participants selected which tie to cut from this list of alters.

If a participant cut a tie, they were then given a list of all participants in the network to whom they were not currently tied, from which to propose a new tie. During this tie addition phase, participants could see prospective alters' total number of monetary units. In the control condition, where political identities were not visible or known, reputation scores followed prior work[4,5,8] and were simply the average number of monetary units a prospective alter donated to each of their neighbors in the previous three rounds. In the three experimental conditions, they could also see each prospective alter's self-identified political affiliation (red for Republicans and blue for Democrats), and the average number of monetary units they gave their ties—either all ties (undifferentiated reputations condition), only ingroup ties (parochial reputations condition), or ingroup and outgroup ties tallied separately (intra/intergroup reputations condition)—in the previous three rounds. These averages represent the alters' reputations (i.e., their level of cooperation in previous rounds). Hence, which reputational information participants saw depended on the experimental condition to which they were assigned (see Table 1). For instance, in the intra/intergroup reputations condition, each prospective alter had two reputation scores—one for how the alter treated their own ingroup members and one for how the alter treated their own outgroup members. In the parochial reputation condition, prospective alters' reputation scores were based solely on how the alter treated their (alter's) own ingroup members. For example, if ego was a Democrat

and a given prospective alter was a Republican, the ingroup score would be the average number of monetary units given to Republicans over the previous three rounds. Screenshots in the SI show how reputation scores were displayed depending on experimental conditions.

From this list of potential alters, the participant selected a new alter to send a tie request to. Prospective alters who received a tie request then decided whether to approve or deny that request. (There were no constraints on the number of tie requests any given participant could receive and, if they chose, accept.) When deciding whether to approve a tie request, prospective alters could see the requesting player's total number of monetary units, the number of monetary units they donated to each of their neighbors in the previous round, and, in the political identity visible conditions, the requesting player's self-identified political orientation. Since ties between participants represented the only opportunities for interactions in the study, following prior work in this literature[5,37,45,65], those whose tie initiations were denied did not have an opportunity to retaliate against the participant who denied the tie invitation.

If, as a result of tie updates, a participant lost all their ties, that participant would be isolated from the network, removed from the interaction phase of the study, and sent to the post-study questionnaire[8,37,45]. The SI presents results for network isolates.

Given that tie updates occurred only four times during the study, and each participant could only elect to add one alter in each phase, this presents a relatively conservative test of differences in network segregation between conditions. Further, note that this design limits broader strategic considerations of network formation[74,75]. That is, following closely related work[8,9,37,56], participants in our studies can benefit from each additional tie formed, as long as that tie is cooperative. Thus, while participants have an incentive to strategically build their ego networks, there is a limited strategic incentive to form or delete ties in order to affect the broader social network and their location in it[75,76].

The study lasted 18 rounds. But to avoid end-game effects, participants were not told this. After completing the 18 rounds, or becoming isolated from the network, participants completed the post-study questionnaire. At the end of the study, participants were paid $2 for passing the comprehension check quiz, $1 for successfully starting the study, and $1 for every 1000 monetary units accrued during the interaction phase.

## Statistical analyses

**A note about network dependence.** Here we discuss our estimation strategy. When we examine behaviors within networks, and behaviors within networks through time, we violate standard assumptions about observations being (conditionally) independent. Network structure and temporal dependence both induce dependence. The cooperative behaviors of participants, for example, are affected by the behavior of their neighbors. We can evaluate this more formally, testing for network dependence among cooperative behaviors. Lee and Ogburn recommend comparing the observed network dependence to a distribution of dependence based on chance[77]. We computed whether there was significant ($p < 0.05$) network dependence among cooperation values for each network round of our data. Figure S1 shows the count of networks (out of the 10 in each condition) that had significant levels of network dependence. By round two we see high levels of network dependence.

Common strategies for adjusting for network dependence include incorporating network structure via covariates or transitioning to non-parametric inference[78]. In the context of non-parametric inference for regressions, it is preferable to permute model residuals[78]. In our case, however, we have a longitudinal structure as well: we observe cooperation and other behaviors through time. Such serial correlation in the outcome can also invalidate inferences. In light

of these differential sources of nesting, we include variance components to adjust for temporal nesting (i.e., serial correlation) while permuting our outcome to adjust for network dependence. Each model description includes a note about the resampling scheme. Because we rely on these nonparametric methods, we do not estimate the standard errors associated with our regression coefficients and instead, base inferences on comparisons of the observed regression coefficients to a distribution of coefficients from resampling on our dependent variables. A consequence of not having standard errors includes the inability to estimate confidence intervals and effect sizes for most outcomes (exceptions include analyses reported in Tables S2 and S11). Supplementary Note 1 of the SI reports a range of sensitivity analyses that illustrate the robustness of the findings reported in the Main Text.

**Cooperation.** All statistical analyses are two-tailed. Figure 1A shows average cooperation rates through time for each experimental condition. Consistent with past work on dynamic networks[4,8], we observe high levels of cooperation at the end of the study. Early on, however, differences in cooperation patterns and corresponding network dynamics may shape aggregate outcomes. To investigate cooperation before equilibrium, we modeled cooperation in rounds 1–8. Table 3 presents three linear mixed models predicting cooperation decisions with each alter. In this specification, alters are nested in participants, and participants are nested in networks. We also included an AR(1) specification to adjust for serial correlation[79]. We predict cooperation as a function of political similarity and experimental conditions in Model 1 and include their interaction in Model 2. Model 3 adjusts for direct reciprocity since direct reciprocity has powerful effects on cooperation[2,44]. We operationalize direct reciprocity as the amount the participant received from their partner in the previous round. As such, all first interactions with partners are missing on this variable. Inference is based on permuting cooperation within network-rounds 1000 times and computing the proportion of permutation-based coefficients that exceed the observed ones in magnitude. Margins for Fig. S2 are drawn from the model with only main effects (Model 1), and margins for Fig. 1B are drawn from the model with the interaction effect included, controlling for direct reciprocity (Model 3). Because our model-based estimates of variance are inconsistent due to nesting[77], we cannot rely on the popular Delta method to estimate the uncertainty associated with marginal cooperation. Instead, we use bootstrapping, sampling 90% of the network-rounds with replacement, computing the margin, and then repeating this 1000 times. The inner 95% of the bootstrapped distribution serves as the bounds of uncertainty in Figs. S2 and 1B.

**Whether to drop an alter.** Figure S3 shows the proportion of participants who opted to drop an alter for each tie deletion phase by experimental condition. Participants in the intra/intergroup reputations condition were least likely to drop an alter. Those in the control condition approached the intra/intergroup condition by the second tie update phase, but then dropped more alters in phases 3 and 4. We modeled these proportions with mixed effects logistic regression, with time in participants, and participants in networks. Inference in this model is based on permuting whether to accept ties within network rounds. As shown in Table S1, after controlling for round, the number of partners, the amount given, the amount received, and the proportion of the participant's network that shares a political identity, we find significant effects for the experimental condition. Figure S4 shows the probability of dropping an alter by condition. The bootstrapped distribution of margins came from sampling 90% of the data with replacement 1000 times. We find that participants in the inter/intra group condition are the least likely to drop an existing alter, those in the parochial condition are the most likely to drop an existing alter, and those in the control and undifferentiated conditions are in between and about equally likely to drop an alter.

**Which ties were severed?.** Conditional on deciding to drop an alter, participants then selected one of their current alters to drop. Figure S5 shows average (A) cooperation and (B) ingroup effects by whether the alter was dropped and the experimental condition. Figure S5A shows that those who were dropped were less cooperative across experimental conditions, and Fig. S5B shows that those who were dropped were less likely to be ingroup members in the undifferentiated and parochial reputation conditions. We model participant decisions to drop an alter as a conditional logistic regression, sometimes called fixed effects logistic regression[80]. We model the selection process as a function of alter endowments, how much the participant received from the alter (i.e., direct reciprocity), the alter's reputation, and whether the alter is homophilous. Results are reported in Table S2. In Model 2, we include a participant-level variable denoting experimental conditions. While the main effect for this term cannot be estimated, the interaction effect with the same party can be, and that tells us how the ingroup effect changes with the condition. Estimates of the ingroup effect from Model 2 generated Fig. 2A. In this case only we rely on normal theory estimation since permuting who the participant dropped does not work (participants had ~4.5 ties, so 1000 permutations of this is not a realistic distribution). We use the sandwich estimator to generate standard errors[81].

We observe the largest ingroup effect in the parochial reputation condition. To illustrate, we computed the probability that a participant would select an outgroup member when choosing between (for simplicity) a single ingroup and a single outgroup member. Figure S6 illustrates the implications of the ingroup effect on dropping alters, showing the marginal probability of dropping the outgroup member at various levels of independent variables. We set their endowment and reputation to the mean, and systematically varied how much a hypothetical ingroup/outgroup pair gave the participant, and computed the probability the participant would select the outgroup member.

**New alter selections.** Once participants decided which of their current alters to drop they were shown a list of all available alters and asked to propose a new tie to one of them. Figure S5 shows (C) average cooperation and (D) the proportion of same party others for tie proposals. Figure S5C shows that those alters who were selected by participants for new ties were more cooperative than those alters who were not selected. Similarly, Fig. S5D shows that selected alters were more likely to share a political party in all our experimental conditions. As with the choice model above, we use conditional logistic regression to model alter selection from the pool of available alters. Here, however, the pool of available alters was large enough to permute so we rely on non-parametric inference. We model alter selection as a function of alter endowments, same-party effects, and reputations. Endowments were $z$-transformed within rounds because they increased over time.

As noted above, the information available to participants varied by experimental condition. Participants in the control and undifferentiated reputations condition saw the average amount potential alters' neighbors received from them. In the parochial condition, participants saw only the average amount given to alters' same-party neighbors. In the intra/intergroup condition, participants saw the average amount alters gave to same-party neighbors and the average amount alters gave to opposite party neighbors. Given that this information varied with condition, we are unable to model these data simultaneously without excluding the variable for reputations. As such, Table S3 presents four conditional logistic regression models, estimated separately for each experimental condition. Each model includes the relevant variable(s) for reputation information for that condition. The marginal effects of same-party others in all three models are depicted in Fig. 2B and were generated using a bootstrapped sample of individual choices (90%, with replacement). Figure S7 illustrates the effect of same party others on alter selection in the

undifferentiated reputations condition. As above, we set endowments to their mean and varied only the relevant reputation score of a single ingroup and a single outgroup member to illustrate the probabilities of selection.

**Accepting tie requests.** As described in the main text, we find that accepting pending tie requests is primarily driven by the requester's reputation. As shown in Table S4, neither the main effect nor any of the interaction terms with experimental conditions are significant for same-party effects.

**Network measures.** As noted in the main text, network clustering is defined as the probability that adjacent nodes are connected[47]. We computed this using the *transitivity* function in the *igraph* package for R[82]. Similarly, segregation is defined as having fewer between group ties than expected by chance given the density of the network[48]. We computed it using the *freeman* function in the *netseg* package for R[83,84].

**Becoming isolated from networks.** Participants could have become excluded from our study for multiple reasons. If participants did not respond to the application within 10 s of being prompted, participants were dropped to avoid entire networks from crashing. This may have happened due to internet connectivity, for example. But, of the 1073 participants who began our study, 855 completed it. Of the 1073 participants who began our study, 139 of them were isolated from experimental networks due to network dynamics. That is, they began a tie update phase with at least one alter and ended that phase with none, isolating them from the network. The SI models network isolation.

After becoming isolated, participants were sent to the post-study questionnaire. Figure S8 shows the count of isolates observed by experimental conditions for each tie update phase. We do not observe any systemic patterns in the Figure. Table S5 presents the results of a Cox proportional hazards model, predicting the hazard of participants becoming isolated from our networks. Due to the possibilities of network dependence shaping our hazard model results, estimates of uncertainty in Table S5 come from permuting the time-relevant variables for participants, holding the network constant again. We find that participants who gave more and who had larger endowments were less likely to become isolated from the networks, but that experimental condition is unrelated to network isolation.

**Missing data.** As noted above, of the 218 participants who did not complete our study, 139 of them became isolated and the remaining 79 dropped out for other reasons (e.g., internet connectivity). If these drop-outs occurred systematically, there might be concerns about sample selection. Table S6 shows how the 79 dropouts were distributed across our experimental conditions. Row 1 shows that the fewest dropouts were in the control and the most dropouts were in the intra/intergroup condition. Importantly, the distribution of dropouts is not significantly different from a uniform distribution ($\chi^2_{(3)} = 6.42$, $p = 0.093$, univariate Chi-squared test). In the control condition, we did not have participant's political affiliation (because this was measured in a *post-study* questionnaire). In the experimental conditions, we asked this question before participants were allocated to networks, so we can assess whether dropping out varies by politics. Table S6 also shows dropouts by political affiliation (except for the control condition). There were 34 Republicans and 34 Democrats who dropped out. While there are some differences by experimental condition, the count of dropouts by political affiliation and the experimental condition is not significant either ($\chi^2_{(3)} = 5.61$, $p = 0.132$, bivariate Chi-squared test).

**Reporting summary**
Further information on research design is available in the Nature Portfolio Reporting Summary linked to this article.

## Data availability
The processed data, along with distributions of regression coefficients that were used for non-parametric inference are available at: https://doi.org/10.17605/osf.io/xkamd. All results, figures, and tables can be reproduced with these data. All raw data are also available upon request from the third author.

## Code availability
The R code to reproduce the statistical results, the code that generates the non-parametric inferential statistics, and the code to generate Figs. 1–4 is also deposited at https://doi.org/10.17605/osf.io/xkamd.

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

## Acknowledgements

This research was supported by grant W911NF-19-910281 from the Army Research Office to the first and third authors.

## Author contributions

Author contributions: B.S. and D.M. designed research; B.M. performed research; D.M. analyzed data; and B.S., B.M., and D.M. wrote the paper.

## Competing interests

The authors declare no competing interests.
