## [Peer Review File · Nature Communications]

Reputations for treatment of outgroup members can prevent the emergence of political segregation in cooperative networksReviewers' Comments:

Reviewer #1:

Remarks to the Author:

In this clever and timely study, the authors find that tracking—and publicizing—how people behave toward members of an opposing political camp, can mitigate polarization's divisive effects. Specifically, they find that "reputation systems" which maintain separate tallies of how people behave toward ingroup and outgroup members result in more cross-cleavage network ties and cross-cleavage cooperation than systems that keep a combined tally or only track behavior towards the ingroup. They use a variety of metrics to show this is the case (e.g. severing ties, adding new ties, cooperating). While not all of their results are statistically significant, most are and the direction of their findings is consistent and their overall preponderance of evidence clearly supports this argument. Therefore, I support publication, provided that the authors 1) foreground certain limitations and scope conditions, 2) improve the clarity of their narrative, and 3) address an important network-related statistical issue discussed below.

Major Points

1) Limitations

- My impression is that all the differences washed out by the end of the game and that we really only see meaningful differences through round 8. In light of this, the authors need to justify why their findings are still important. Is the key that the networks look different at baseline even though behavior doesn't? If so, make that more clear early on.

- Players are after money, not popularity or kudos, so their only real incentive to mistreat outgroup members is fear of being betrayed. Betrayal also has costs—the other player may stop trading points with you for the remaining rounds before getting a chance to switch ties. Given that this game is incentivized by money (not popularity or prestige), I would not expect players to mistreat the outgroup as a way to curry favor with their co-partisans. In contrast, Twitter users really might acquire more likes or retweets from mistreating outgroup members, and there are no clear costs to doing so. This difference in incentive structures might lead to different results. My point is not that Twitter should be the default point of comparison, but rather that the authors should discuss what sort of real-world systems their experiment is most likely to generalize to and where it is less likely to apply.

2) Clarity

- Despite being familiar with the network experiment literature, I didn't really understand the experiment until p4, paragraph 2. Specifically, I missed what the difference was between "colorblind" and "control." In that paragraph, however, the authors first contrast control with all three treatment conditions and then in the next sentence differentiate among the three treatments. That was what made it click for me. I suggest the authors adopt a similar structure the first time they explain the experiment in p2 paragraph 5, or even in the abstract

- It was not clear until I read the SI that a "reputation score" was literally the number of MUs awarded. I began to suspect this near the end of p5, but by that time I already had an image in my head of players giving one another endorsements or ratings. After all, "reputation" is often used to mean what people think of and say about you, regardless of whether that opinion is deserved. And in "reputation systems" like Yelp or Amazon, reputation is an aggregation of customer ratings, not dollars spent. Therefore, I think it is important to make this crystal clear early on in the Empirical Strategy section, perhaps at the start of the explanation of reputation.

- After reading the abstract, I was unclear as to what the major findings were. The three treatment conditions come off as equally important, so it is not at all clear until well into the paper that the inter/intragroup condition is the focus of the article and the source of the most exciting findings. It's hard to convey such a complex experiment in so few words, so some simplification may be needed. One way to do this would be to set up a problem and then propose a solution. For instance, "As predicted, when players were told their neighbors' political affiliation, they showed favoritism toward co-partisans. However, we were able to reduce such polarization by revealing to everyone how players had behaved towards both their own party and the other side." I would minimize the amount of specific terminology like "parochial reputation system" or "clustered on cooperation" and just focus on getting the main idea across. For instance, the sentence that begins "In contrast" could just say "...resulted in networks in more cross-party cooperation and network ties." without having to list the specific metrics, all of which will be opaque to someone who has yet to read the article.

- It would have been helpful to know sooner that the study participants are playing a game at all. The authors could add a clause mentioning that participants are playing iterated prisoner's dilemma on p3 or 4, rather than waiting to p5. That will make the empirical strategy easier to follow.

- Homophily is a bit of an ambiguous term in the network literature, but I tend to see it used as a cause, not an outcome. That is, a Democrat's desire to have friends who are Democrats is homophily, whereas a tie between two Democrats is merely the *result* of homophily (or of some other process). The authors use the term both ways and this creates confusion. I would suggest sticking to homophily as an explanation/cause, and referring to ties between co-partisans as "in-group ties" or even just "same party" in the tables and figures.

3) Statistical issues

- Regression in a network setting is not straightforward. Although the randomness of treatment ensures that estimates are unbiased, I am concerned that the authors are underestimating the variance. Observations in network experiments are not independent. We should expect the respondents' outcomes to be correlated with those of their neighbors. Moreover, they may also be correlated with those of non-neighbors, since their behavior every 4th round takes the behavior of all other players into account. I am not entirely sure what the correct statistical approach here is—perhaps clustering standard errors at the network level. Whatever it is (and even if the current approach is valid) the authors need to address it. Many readers who have worked with networks will be hyper-aware that networks create these sorts of complications and will be looking to see how the authors deal with them

- Relatedly, I don't see any mention of autocorrelation across rounds. Player A's outcome in round 4 will likely be correlated with their outcome in round 8. HAC-robust standard errors might be a way to deal with this, though I'm uncertain if it is possible to simultaneously address the autocorrelation problem and the peer effects problem just discussed

Minor points

- Somewhere in the paper or appendix, it would be useful to see the reasoning behind some of the design choices. At present, I can't tell which choices were made for simplicity's sake and which had a theoretical motivation for instance:

- why not allow respondents to drop or add ties every round?

- why reveal respondents' reputations every 4th round rather than constantly updating?

- why limit how many ties they can drop or add?
- why set up the experiment in such a way that the networks will almost inevitably become more sparse over time?
- It would be helpful to remind the reader more than once that "cooperation rates" are literally the average number of MUs shared, not the % of the time players share more any MUs or the % of players who do so. For instance, add this to the y-axis label or the caption in Figure 1 and mention it somewhere in the discussion a few paragraphs after it was first introduced. It's not intuitive so it's easy to forget. Or better yet, be explicit in the text and refer to it as "average number of MUs shared" or something to that effect
- I find it a bit surprising that cooperation in the inter/intragroup condition starts out higher and stays higher by about the same amount throughout the 16 rounds (see Figure 1A). That is, respondents must have realized from the outset that it behooves them to be nice to the outgroup; it's not something they learned through trial and error. The implications of this finding may be quite interesting, though if the authors don't have space to discuss it in the paper, that's fine.
- Players were given a fairly complicated task. Personally, I found the instructions on SI p15 confusing and might have failed the comprehension check even if I was trying to pay attention. Hence, it would be good to know what % failed the comprehension check and whether the failure rate differed by condition.
- Does the game's complexity limit its generalizability? Might the average American be less strategic—or less likely to understand their incentives—than the average player who passed all the comprehension checks (or liked games enough to stay in study once they saw the instructions)? Would that bias the results upward or downward?
- SI p13 should say "blue" instead of "red" I think
- Could some of these homophily effects (or what I would call "in-group tie" effects) be driven by one party simply less homophilous? More cooperative? More trusting of the outgroup? If that's a possibility, then control for party (or explain in response memo why doing so would be problematic)
- Maybe this is unanswerable, but it would be interesting to tell readers how much of the improvement in cooperation over time is due to a) players starting to behave better toward everyone, b) loss of ties between people who don't trust each other, or c) isolation/elimination of serial defectors. In other words, is this mainly a story about behavior improving or about networks improving? This could help address the first point I raised about Limitations
- The reference category is "control" in Table S1 but switches to "colorblind" in Table S2. Why?
- In Table S4 does "Average given to ingroup" mean "average that the person I might form a tie with gave to their own ingroup" or "average that the person I might form a tie with gave to *my* ingroup"? Same for outgroup.
- Table S4 would be easier to understand if control and colorblind were separate models. As is, the need for that interaction term just makes it complicated.
- p8 "In parochial reputation systems, participants' reputations were determined solely by how they treated fellow ingroup members. In a highly polarized environment, where ties to outgroup members are rare, parochial reputation systems are arguably the default: at the extreme, it becomes difficult to develop a reputation for treating outgroup members cooperatively, since one rarely interacts with

them, and information flows primarily through clusters of ties between ingroup members" In that case, wouldn't the default be knowing next to nothing about outgroup member's reputations? That's a bit different than the parochial system here, where you have complete information about how each outgroup member treats their own group

- p9: "These findings provide insight into the effective design of reputation systems when populations risk segregating into clusters based on salient group identities" Great point. Perhaps the authors could hint at this in the intro or even abstract to help motivate the paper (this could also be a way to address the first limitation point I raised)

Nature Communications encourage referees to sign their reports, so I will so do. Assuming this goes to R&R, which it ought to, you are welcome to reach out to me for clarification on any points if needed.

Sincerely,
Matthew Simonson
Postdoctoral Fellow, Political Science
University of Pennsylvania
(and perhaps more relevantly, graduate of Network Science Ph.D. program at Northeastern University)

Reviewer #2:

Remarks to the Author:

The manuscript investigates the important question of how cooperation can be sustained by a reputation-based mechanism in large social networks characterized by political differences. Using a large web-based experiment with endogenously changing network ties, the authors examined how cooperation and segregation by political attitudes emerge depending on the choice of reputation systems applied: whether reputations track behavior of outgroup members, whether reputations are colorblind, or whether reputations track ingroup members only. The results of the experiment are counter-intuitive, as they show that higher cooperation levels and more clustering on cooperation rather than on political opinions are achieved if reputations are distinguished between treatment of ingroup and outgroup members.

The questions are highly important, and the results are noteworthy. The results could be of great significance to the study of cooperation as well as political segregation.

Unfortunately, there are some fundamental problems with the experimental design, with the analysis, and with the interpretation of the results.

Introduction

The questions that are addressed in this study are not precisely formulated. It is clear that overall cooperation, segregation by action, network dynamics, and political segregation are the outcome variables, but the research questions formulated do not properly reflect on the manipulations carried out in the experiment.

Controlled experiments, by nature, could investigate only a limited set of research questions and a limited set of hypotheses. It depends on the number of manipulations and experimental conditions how many exactly. The manuscript introduces many questions, much more than what can actually be checked in an experiment, and it is not clear in the introduction what exactly is going to be tested. It does not become clear what is meant by reputations and reputation systems. It is somewhat misleading to use the word reputation, because it is not a subjective category formed by individuals about others in the experiment, but an objectively communicated image score (cf. Nowak and Milinski). As this is not clear from the beginning, the reader is a bit confused and has the false impression that the study would be about statistical discrimination, conditional cooperation, or parochial cooperation.

It is also less clear what the expectations and hypotheses are concerning the main manipulation: the "type of reputations". These are in fact different image scoring systems that are tested against each other. Individuals have no choice on forming reputations and the system settings apply for all subjects in a given experimental session. Therefore, it is highly misleading to talk about "a person's treatment of ingroup and outgroup members", "reputational information is more apt to flow between ingroup members", and different kinds of reputations as if they were individually made. Image score systems are imposed in the experiment instead for testing the impact of image scores under different information settings (universal, parochial). Concerning these, as you say, "competing arguments" are tested, but it is not argued clearly on which hypotheses theoretical perspectives would agree upon and on which they would have diverging expectations. What is the "best case" scenario? High cooperation and no political segregation? What about massive exclusions (isolation)? A very innovative element of the theorization and of the experiment is the exploration of retaliation against outgroup members. Although the concept of intergroup conflict falls far from the design and such references are over-generalizations, this is still a very interesting question and could be justified by various streams of literature.

Inconsistencies in the design

Papers cited by the literature review typically consider binary cooperation actions and binary reputations (good or bad) – and it would have been easier to remain at these simple settings to test fundamental theoretical hypotheses on reputation-based cooperation and political segregation.

Cooperation, however, is not binary, but continuous in the experiment.

The presented design considers endogenous network dynamics. Why did you need network dynamics at all to test questions on conditional cooperation? What you have is certainly interesting concerning network dynamics, but this comes with the loss of control in the design and complications for individual-level analysis (for some methodological solutions, see for instance, Ule, A. (2008). *Partner choice and cooperation in networks: Theory and experimental evidence* (Vol. 598). Springer Science & Business Media.).

Tie formation (unlike tie deletion) is regularly assumed to require mutual consent, which creates some difficulty for experimental designs. The current design is also based on mutual consent for tie formation, but with a limit of a total of four opportunities to cut and add ties. (But how did the max. 4 determine which choices are executed at the end? What was the exact algorithm, for instance if somebody received 20 requests?) A further complication for dynamic network experiments is that information is needed to make a justified choice for establishing a new tie. The details of how these difficulties are handled in the experiment are not explained. Moreover, it is not clear how tie deletion and formation phases followed each other.

What is meant by "total number of points"? Total MUs earned? (This is trivial, but it is not explained in the text). For the "reputation system" manipulations, a table would help to clarify the design differences.

Subjects were not presented with the full network of colored nodes (this is the impression of the reader from the text), but in fact they knew their network neighborhood only (Supp. Mat.). This is not clear from the text (especially concerning tie formation). A more precise description of what information was available would be essential to understand decisions of tie formation.

It seems that the network size has not been fixed between sessions. This creates a problem for the internal validity of the results. Furthermore, random graphs have the problem of differences in individual structures at the outset, hence individual decisions are no longer comparable. Why didn't you start from a regular graph? The proportion and distribution of Democrats and Republicans in the network could also make a difference. Although this could be balanced out with a very large N, but statistical control would still be necessary to rule out effects of distortions in the (structural) distributions.

Given the strategic nature of tie deletion and tie formation, I suspect that insisting tie initiations could also been acknowledged by conditional action and tie deletions by the partner could result in some frustration and retaliation or generalized (indirect) retaliation.

Limitations

There are some natural limitations of the study that need to be emphasized.

There is no major difference in the observed cooperation rates across image score system conditions (Fig. 1A). A lot of statements still emphasize differences, which are over-interpretations. Repeating the game and the availability of endogenous partner choice had a much larger impact on cooperation in every condition. There is a natural tendency of defectors becoming isolated, which drives cooperation in repeated social dilemma games with partner choice or exclusion. Furthermore, it seems that objective image scores work in general and enable a high level of cooperation in any condition. A multilevel model on individual decisions with appropriate controls could be part of the main text. To what extent could individuals make statistical generalizations about average group reputations and their variance? Is there a tendency for statistical discrimination?

A speculation on whether Democrats (or Republicans) had a status advantage is missing. Based on status characteristics theory or based on double standards, Democrats might have an advantage of being perceived as more cooperative already at the outset. There might be double standards with retaliation as well as it could be more inappropriate for a Democrat to be a free rider than for a Republican.

Complications arise with possible concerns of second and higher order norms. Defections might be considered right if they target an individual with low reputation (or an out-group member). The design unfortunately cannot exclude the complications arising from justified punishment and other higher order social norms.

Reviewer #3:

Remarks to the Author:

This paper investigates how reputation systems interact with group identities in the formation of networks and cooperation. In particular, the authors study network dynamics among polarized groups (democrats vs republicans) across three experimental treatments: parochial reputations, colorblind reputations, and inter/intragroup reputations. The authors find that people are more cooperative when they can learn the reputations of both ingroup and outgroup members, and that such networks show also less segregation between political groups.

I think this is an excellent study that provides important contributions for both theory and practice. As the authors mention, they find compelling evidence that reputation systems are unbounded, a result that does not support social identity perspectives but rather recent findings that highlight the importance of reputation in increasing cooperation with outgroup members. At the same time, the authors also offer a powerful tool (reputation systems), that can be used to decrease political polarization. Moreover, I find the statistical analyses sound, and the paper well written.

I have a few suggestions that I think might enrich the paper and clarify some aspects of the methodology.

Individual heterogeneity. Previous research found that political ideology moderates parochial cooperation (Balliet, et al, 2018, Journal of Conflict Resolution; Romano et al., 2021 Phil Trans B). I was wondering whether the authors could check whether they find systematic differences between democrats and republicans in how they deal with and react to reputation systems. I understand that the authors have a majority of democrats, but I guess it would still be important to discuss such potential differences. Someone might claim that these results (e.g., the positive effect of the intra/intergroup network) are driven by the fact that the majority is democrat. Can the authors address this potential concern with the data? If not, I would recommend to include this as a potential issue/limitation in the discussion section.

Extreme cases. Relatedly, how are people in the extreme of the political ideology scale behaving compared to more moderate people? I guess that to defend social identity approaches, one might argue that the inter/intragroup finding is driven by people that responded 3 or 4 (hence the middle

points) in the initial political ideology question. Also in this case, I would suggest the authors to check whether they can address this potential issue with additional data analyses, or mention this as a potential limitation in the discussion section.

Methodology/design/sample. I think that the authors could provide more details about the data-collection and methodology. I find remarkable that the authors were able to collect such amount of data for real time interactions across 18 rounds. With such large sessions, I find difficult to believe that the authors did not encounter drop outs in their data collection. Maybe I missed this, but I would suggest to report any drop out rate encountered during the sessions, and discuss how this might have potentially affected the results.

Reviewer #4:

Remarks to the Author:

The goal of this paper is to empirically test the effect of identity-based reputations – here political identity – on network formation and cooperative behavior within these networks. The empirical analysis seems to be very well done and the conclusions are clear: allowing users to develop reputations in terms of how they treat out group members leads to more cooperative behavior and more inclusion of outgroup members within networks than reputations that are either based only on how one treats ingroup members or are “colorblind”, in the sense that the reputations do not distinguish between how users treat outgroup members from ingroup members.

My sense is that the research is interesting and well executed, and certainly speaks to a very important problem in contemporary US society – the tendency of political polarization to drive people into networks that are increasingly segregated by ideology, with the assumption that this will in turn lead to further polarization. I think it therefore is certainly worth consideration by the editors for publication in Nature Communication, but I want to be very clear that I am not a network scientist, so I can not speak to the quality or the appropriateness of the network methods employed by the authors. From that perspective, though, I would like to offer three suggestions that I think would help make the paper stronger – and have more of a contribution to a wider audience – in a revised version of the manuscript.

First, the paper is at the same time both dense and a bit repetitive. I had to read the manuscript several times before I understood the distinction between the three different reputations mechanisms and the various tests of the hypotheses. I think there is a way to do all of this in a much more streamlined way, perhaps involving a table that shows the relationship between the different reputation mechanisms and the various dependent variables, showing exactly what is predicted in each case.

Relatedly, were these hypotheses pre-registered? Apologies if I missed it, but I could not find anything in the text about pre-registration.

Second, and perhaps most importantly, I think the paper would benefit from a deeper discussion of the external validity of findings related to a game where there are incredibly clear benefits from cooperation. As the authors note, everyone figures this out eventually and cooperation goes up over rounds. So the question is, what does this capture about the motivating example of the paper: the seemingly unstoppable rise of political polarization in the United States? What is the game everyone is playing where there are clear benefits from cooperation? If anything, from a political science perspective, we tend to assume that political competitions are zero-sum: if my side gets control of the government and implements my preferred policy outcome, then by definition the other side does not get control of the government and can not implement their preferred policy outcome. I am willing to accept the results of the experiments as confirming the authors’ hypotheses, but I think the potential impact of the paper would be much larger if the authors could make a compelling argument about

what system they are actually capturing with their experiment. In other words, are we learning anything outside the confines of the experimental treatments?

Third – and related to the previous point -- what are real world institutions that could privilege these types of reputations? This is where the paper ends up: arguing for institutions that privilege reputation building that distinguish between how one treats in groups and out groups, and against reputations that are color blind or focused on how one treats only one's own group. Fair enough, but what are examples of such institutions, including both institutions to be avoided and institutions to be recommended? It does seem to me that there is an interesting opportunity here for the authors to discuss the differences between offline and online networks. Off the top of my head, my sense would be that these types of institutions would be easier to design in online networks – where platforms can tinker with the rules of the game – than they are in offline networks, but either way I think it would behoove the authors to offer their thoughts in this regard.

Minor point: I would recommend labeling the units on the Y-Axis in the figures. For example, in Figure 1 make it clear that these are the monetary units (at first glances, it looks like it could be percentages).

We greatly appreciate the opportunity to revise and resubmit our manuscript to *Nature Communications*. The comments and criticisms of the four reviewers were among the most helpful we have seen and the changes we made based on these comments have produced a far better manuscript. Below, we reproduce the Reviewer comments and respond to them in indented, italicized text.

Following Nature Communications policy, we have submitted both a “clean” version of the revised document and a “tracked changes” version that highlights all changes since the initial submission. So that it is clear where we made revisions, *the page numbers below refer to the tracked changes document.*

Reviewer Comments

Reviewer #1 (Remarks to the Author):

In this clever and timely study, the authors find that tracking—and publicizing—how people behave toward members of an opposing political camp, can mitigate polarization’s divisive effects. Specifically, they find that “reputation systems” which maintain separate tallies of how people behave toward ingroup and outgroup members result in more cross-cleavage network ties and cross-cleavage cooperation than systems that keep a combined tally or only track behavior towards the ingroup. They use a variety of metrics to show this is the case (e.g. severing ties, adding new ties, cooperating). While not all of their results are statistically significant, most are and the direction of their findings is consistent and their overall preponderance of evidence clearly supports this argument. Therefore, I support publication, provided that the authors 1) foreground certain limitations and scope conditions, 2) improve the clarity of their narrative, and 3) address an important network-related statistical issue discussed below.

Thanks for the kind comments. We agree with these suggestions and detail those we made in response to your specific comments and suggestions below.

Major Points

1) Limitations

- My impression is that all the differences washed out by the end of the game and that we really only see meaningful differences through round 8. In light of this, the authors need to justify why their findings are still important. Is the key that the networks look different at endline even though behavior doesn’t? If so, make that more clear early on.

We agree we were unclear in the previous submission, as also indicated by a very similar question from Reviewer 2. We now make clear early on (see p. 3) that given the well-known power of dynamic networks, we expected that any condition-level differences in cooperation would dissipate after participants had opportunities to shed ties and select new ones. Critically, however, we aimed to show that these early differences in cooperation with ingroup vs. outgroup members would lead to condition-level differences in network structure. We reiterate these points at other points in the manuscript.

- Players are after money, not popularity or kudos, so their only real incentive to mistreat outgroup members is fear of being betrayed. Betrayal also has costs—the other player may stop trading points with you for the remaining rounds before getting a chance to switch ties. Given that this game is incentivized by money (not popularity or prestige), I would not expect players to mistreat the outgroup as a way to curry favor with their co-partisans. In contrast, Twitter users really might acquire more likes or retweets from mistreating outgroup members, and there are no clear costs to doing so. This difference in incentive structures might lead to different results. My point is not that Twitter should be the default point of comparison, but rather that the authors should discuss what sort of real-world systems their experiment is most likely to generalize to and where it is less likely to apply.

This is an excellent point and relates closely to one made by Reviewer 4. We now address it at length in the Discussion, which we have extensively rewritten to address this and other issues raised in the reviews. See, in particular, pp 13-14.

2) Clarity

- Despite being familiar with the network experiment literature, I didn't really understand the experiment until p4, paragraph 2. Specifically, I missed what the difference was between "colorblind" and "control." In that paragraph, however, the authors first contrast control with all three treatment conditions and then in the next sentence differentiate among the three treatments. That was what made it click for me. I suggest the authors adopt a similar structure the first time they explain the experiment in p2 paragraph 5, or even in the abstract

This issue was also raised by Reviewer 2. We did not clearly differentiate between conditions early enough in the previous draft and have taken steps to clearly do so in the revised paper, both on pp. 3-4 and in the abstract. Also, following Reviewer 2 and 4's suggestions, we have added Table 1, which includes (among other details) a brief description of the conditions, which should further help clarify differences between conditions.

- It was not clear until I read the SI that a "reputation score" was literally the number of MUs awarded. I began to suspect this near the end of p5, but by that time I already had an image in my head of players giving one another endorsements or ratings. After all, "reputation" is often used to mean what people think of and say about you, regardless of whether that opinion is deserved. And in "reputation systems" like Yelp or Amazon, reputation is an aggregation of customer ratings, not dollars spent. Therefore, I think it is important to make this crystal clear early on in the Empirical Strategy section, perhaps at the start of the explanation of reputation.

Great suggestion. We now explain how we operationalize reputations in the second paragraph of the Empirical Strategy section (see p. 5) and note the precedent for this operationalization in existing literature. We also revisit this at several other points in the paper, including in the Discussion (where we call for future research on alternative operationalizations of reputations).

- After reading the abstract, I was unclear as to what the major findings were. The three treatment conditions come off as equally important, so it is not at all clear until well into the paper that the inter/intragroup condition is the focus of the article and the source of the most exciting findings. It's hard to convey such a complex experiment in so few words, so some

simplification may be needed. One way to do this would be to set up a problem and then propose a solution. For instance, "As predicted, when players were told their neighbors' political affiliation, they showed favoritism toward co-partisans. However, we were able to reduce such polarization by revealing to everyone how players had behaved towards both their own party and the other side." I would minimize the amount of specific terminology like "parochial reputation system" or "clustered on cooperation" and just focus on getting the main idea across. For instance, the sentence that begins "In contrast" could just say "...resulted in networks in more cross-party cooperation and network ties." without having to list the specific metrics, all of which will be opaque to someone who has yet to read the article.

Thanks for this suggestion. This is a really great approach to communicating our key findings quickly and effectively. We have included the suggested text almost verbatim in the revised abstract.

- It would have been helpful to know sooner that the study participants are playing a game at all. The authors could add a clause mentioning that participants are playing iterated prisoner's dilemma on p3 or 4, rather than waiting to p5. That will make the empirical strategy easier to follow.

Again, good point. We now state early on – in the abstract and again on page 3 – that ties represented the opportunity to make decisions in an iterated PD.

- Homophily is a bit of an ambiguous term in the network literature, but I tend to see it used as a cause, not an outcome. That is, a Democrat's desire to have friends who are Democrats is homophily, whereas a tie between two Democrats is merely the *result* of homophily (or of some other process). The authors use the term both ways and this creates confusion. I would suggest sticking to homophily as an explanation/cause, and referring to ties between co-partisans as "in-group ties" or even just "same party" in the tables and figures.

We appreciate the suggestion and have made changes throughout so that when we refer to homophily, we are referring to a cause/explanation, rather than an outcome. We agree it is clearer.

3) Statistical issues

- Regression in a network setting is not straightforward. Although the randomness of treatment ensures that estimates are unbiased, I am concerned that the authors are underestimating the variance. Observations in network experiments are not independent. We should expect the respondents' outcomes to be correlated with those of their neighbors. Moreover, they may also be correlated with those of non-neighbors, since their behavior every 4th round takes the behavior of all other players into account. I am not entirely sure what the correct statistical approach here is—perhaps clustering standard errors at the network level. Whatever it is (and even if the current approach is valid) the authors need to address it. Many readers who have worked with networks will be hyper-aware that networks create these sorts of complications and will be looking to see how the authors deal with them

- Relatedly, I don't see any mention of autocorrelation across rounds. Player A's outcome in round 4 will likely be correlated with their outcome in round 8. HAC-robust standard errors

might be a way to deal with this, though I'm uncertain if it is possible to simultaneously address the autocorrelation problem and the peer effects problem just discussed

These are extremely useful comments and suggestions. We revisited the literature on statistical inference with networks. As you correctly note, we have multiple sources of nesting - people in networks and time in people. While the temporal structure and its corresponding variance could be captured parametrically, we could not identify any model or modeling framework that fits our network data structure. In such situations, Snijders (2010) recommends one of two things: control for structure via covariates or use non-parametric inference. Controlling for structure is ambiguous in that the number of possible network statistics is large, leaving open the possibility of omitted network variable bias. This is why we adopted a non-parametric approach. In all our analyses – except one - we relied on permutation tests to estimate the variance in our parameters. In the exceptional case, which we describe in detail in the SI, we relied on robust standard errors. This exception is the conditional logistic regression model for dropping alters. We could not define a reasonable unit on which to define permutations in this case, since participants did not have enough partners to permute this variable.

With respect to temporal dependencies, in the context of our models of cooperation we now include an AR(1) serial correlation specification. The permutation-based results included this term in the model. For our other outcomes, we include time as a fixed effect.

Importantly, these revised estimates of variance did not alter our substantive conclusions.

Minor points

- Somewhere in the paper or appendix, it would be useful to see the reasoning behind some of the design choices. At present, I can't tell which choices were made for simplicity's sake and which had a theoretical motivation for instance:

Good points. We respond to specific design choices in the comments to follow. But we have also addressed this point more generally, and in response to the other comments by you and the other reviewers. The revised paper and revised SI now explicitly motivate our design based on theory and precedent. We appreciate you nudging us to make the rationale for our design choices more explicit.

- why not allow respondents to drop or add ties every round?

As we now explain when we discuss tie updates in the detailed experimental procedures in the SI (see p. 12), we note that this level of tie updates is in the "Goldilocks" range, identified by Shirado et al. (2013). As they note "When the rate of change in social ties is too low, subjects choose to have many ties, even if they attach to defectors. When the rate is too high, cooperators cannot detach from defectors as much as defectors re-attach and, hence, subjects resort to behavioural reciprocity and switch their behaviour to defection. Optimal levels of cooperation are achieved at intermediate levels of change in social ties." Since Shirado et al, studies of cooperation in dynamic networks have typically had tie updates equivalent or similar to ours. Additionally, Wang et al. (2012) demonstrate that allowing one tie update every three rounds results in greater levels of

cooperation (similar to Shirado et al.). We also note in the SI (p. 12) that we chose to allow one tie update every four rounds, rather than three. Our experiment consisted of 18 rounds, and we did not want the final (18th) round to be a tie update with no opportunity for participants to interact with their newly established ties.

- why reveal respondents' reputations every 4th round rather than constantly updating?

This comment could refer to the reputations of those to whom ego is not (yet) connected. If so, our previous comment addresses it. Or it could refer to how much ego's alters contribute to their partners (which is only shown in tie update rounds). We now make clear in the SI (p. 12) that, following previous work (Shirado et al. 2013; Harrell et al. 2018; Melamed et al. 2018), participants were not given information on what their partners' gave to others during the typical decision making rounds since doing so would have likely resulted in information overload (even more so for those who were connected to many alters).

- why limit how many ties they can drop or add?

We addressed this comment in the same text as the closely related comment above about why we did not allow tie updates every round. See p. 12 of the SI.

- why set up the experiment in such a way that the networks will almost inevitably become more sparse over time?

Here, we assume the idea is that since ties can be unilaterally severed but not unilaterally added, this would likely lead to any given person's network becoming smaller over time. We see now that we did not justify this decision and now do so on p. 7 of the manuscript, noting that it is consistent with closely prior work and, perhaps more importantly, consistent with classic sociological conceptions of ties (not to mention lay intuitions of social ties): most relationships can be ended by one person, but require two people to form.

- It would be helpful to remind the reader more than once that "cooperation rates" are literally the average number of MUs shared, not the % of the time players share more any MUs or the % of players who do so. For instance, add this to the y-axis label or the caption in Figure 1 and mention it somewhere in the discussion a few paragraphs after it was first introduced. It's not intuitive so it's easy to forget. Or better yet, be explicit in the text and refer to it as "average number of MUs shared" or something to that effect.

Good point. We now do this multiple times throughout the manuscript (see, e.g., p. 5 and p. 7) We have also added this in the Figure 1 legend and, as suggested, changed the y-axis in Figure 1. We also include this in Table 1, which summarizes each experimental condition and how reputation scores are operationalized for each condition.

- I find it a bit surprising that cooperation in the inter/intragroup condition starts out higher and stays higher by about the same amount throughout the 16 rounds (see Figure 1A). That is, respondents must have realized from the outset that it behooves them to be nice to the outgroup; it's not something they learned through trial and error. The implications of this finding may be quite interesting, though if the authors don't have space to discuss it in the paper, that's fine.

Nice suggestion. We now highlight this finding in the Discussion section (p. 12) and then follow up with some implications later in the Discussion.

- Players were given a fairly complicated task. Personally, I found the instructions on SI p15 confusing and might have failed the comprehension check even if I was trying to pay attention. Hence, it would be good to know what % failed the comprehension check and whether the failure rate differed by condition.

Unfortunately, consistent with previous studies using this platform (e.g., Shirado and Christakis 2017), we did not record the number of comprehension checks people missed. In retrospect, of course, we wish we would have recorded the precise number who failed the comprehension checks. But the primary concern, as you suggest, would be if there were differences by condition. We suspect that this is not the case since the instructions were very similar across conditions. Further, the questions and answer categories were the same across conditions – only one correct answer varied by condition, which was what the reputation score represented.

- Does the game's complexity limit its generalizability? Might the average American be less strategic—or less likely to understand their incentives—than the average player who passed all the comprehension checks (or liked games enough to stay in study once they saw the instructions)? Would that bias the results upward or downward?

Good question. We draw on standard procedures (Shirado and Christakis 2017) and we have refined our instructions over the course of several projects (Melamed et al. 2018; 2022; Harrell et al. 2018). The current study used very similar instructions, changing primarily what was necessary to implement the specifics of this study (e.g., color coding for politics and which behaviors determine a person's reputation in each condition). Further, prior work (now cited in the new discussion on p. 14) shows that behavior in PDs is predictive of parallel behaviors in real life and the most complicated aspect of our design is arguably the choices in the PD. Nevertheless, we now raise this issue in the Discussion (p. 14) and note that future research should assess the generalizability of our results, given these issues.

- SI p13 should say “blue” instead of “red” I think

We have corrected this typo. Thank you.

- Could some of these homophily effects (or what I would call “in-group tie” effects) be driven by one party simply less homophilous? More cooperative? More trusting of the outgroup? If that's a possibility, then control for party (or explain in response memo why doing so would be problematic)

Great questions. Reviewers 2 and 3 raised similar issues. In light of these comments, we added an extensive new section called “Sensitivity Analyses” to our Supporting Information. For every outcome we modeled at the individual level - cooperation, whether to drop someone, who to drop, who to pick, whether to accept, and whether they became isolated - we assessed whether our results were moderated by the political identity of the participant (Republican or Democrat), by whether the participant was politically extreme (extreme [1 or 6] or not [2-5]), or participant sex. In all cases, the

main results hold. We do find a few cases where extremes have amplified effects, but the general patterns and trends are the same.

- Maybe this is unanswerable, but it would be interesting to tell readers how much of the improvement in cooperation over time is due to a) players starting to behave better toward everyone, b) loss of ties between people who don't trust each other, or c) isolation/elimination of serial defectors. In other words, is this mainly a story about behavior improving or about networks improving? This could help address the first point I raised about Limitations

Again, nice question. We suspect that the answer is "all of the above," but we are not aware of any unifying modeling strategy we could use to clearly answer the question. To our knowledge, the closest empirical strategy would be an agent-based model that varied the governing dynamics of interactions and network dynamics. But this approach would rely on simplifying assumptions (of the agents' behaviors) and would likely yield equivocal results since each component (behavior change versus network change) occurred empirically in our results. After giving it much thought, we are not optimistic that this strategy would yield a lot of insight. But we would be happy to introduce such a simulation if the Editor and the Reviewers think it would add significantly to our paper.

- The reference category is "control" in Table S1 but switches to "colorblind" in Table S2. Why?
We have updated the Tables so that the "control" condition is the reference throughout.

- In Table S4 does "Average given to ingroup" mean "average that the person I might form a tie with gave to their own ingroup" or "average that the person I might form a tie with gave to *my* ingroup"? Same for outgroup.

Good catch. We mean "average the prospective alter gave to their own ingroup" and "average the prospective alter gave to their outgroup." This is now clear in the Table.

- Table S4 would be easier to understand if control and colorblind were separate models. As is, the need for that interaction term just makes it complicated.

We agree. The revised table (now Table S3) is easier to understand.

- p8 "In parochial reputation systems, participants' reputations were determined solely by how they treated fellow ingroup members. In a highly polarized environment, where ties to outgroup members are rare, parochial reputation systems are arguably the default: at the extreme, it becomes difficult to develop a reputation for treating outgroup members cooperatively, since one rarely interacts with them, and information flows primarily through clusters of ties between ingroup members" In that case, wouldn't the default be knowing next to nothing about outgroup member's reputations? That's a bit different than the parochial system here, where you have complete information about how each outgroup member treats their own group

This is exactly right. The text did not fit our condition. We have therefore deleted it.

- p9: "These findings provide insight into the effective design of reputation systems when populations risk segregating into clusters based on salient group identities" Great point. Perhaps the authors could hint at this in the intro or even abstract to help motivate the paper (this could also be a way to address the first limitation point I raised);

Nice suggestion. We now anticipate this point in the Introduction (on p. 3).

Nature Communications encourage referees to sign their reports, so I will so do. Assuming this goes to R&R, which it ought to, you are welcome to reach out to me for clarification on any points if needed.

Sincerely,
Matthew Simonson
Postdoctoral Fellow, Political Science
University of Pennsylvania
(and perhaps more relevantly, graduate of Network Science Ph.D. program at Northeastern University)

Reviewer #2 (Remarks to the Author):

The manuscript investigates the important question of how cooperation can be sustained by a reputation-based mechanism in large social networks characterized by political differences. Using a large web-based experiment with endogenously changing network ties, the authors examined how cooperation and segregation by political attitudes emerge depending on the choice of reputation systems applied: whether reputations track behavior of outgroup members, whether reputations are colorblind, or whether reputations track ingroup members only. The results of the experiment are counter-intuitive, as they show that higher cooperation levels and more clustering on cooperation rather than on political opinions are achieved if reputations are distinguished between treatment of ingroup and outgroup members.

The questions are highly important, and the results are noteworthy. The results could be of great significance to the study of cooperation as well as political segregation.

Unfortunately, there are some fundamental problems with the experimental design, with the analysis, and with the interpretation of the results.

We appreciate the concise summary of -- and kind comments on -- our manuscript!

Introduction

The questions that are addressed in this study are not precisely formulated. It is clear that overall cooperation, segregation by action, network dynamics, and political segregation are the outcome variables, but the research questions formulated do not properly reflect on the manipulations carried out in the experiment.

Controlled experiments, by nature, could investigate only a limited set of research questions and a limited set of hypotheses. It depends on the number of manipulations and experimental conditions how many exactly. The manuscript introduces many questions, much more than what can actually be checked in an experiment, and it is not clear in the introduction what exactly is going to be tested.

We appreciate you pushing us on this point. We think our revisions to the introduction and elsewhere – namely in response to your other concerns, and the concerns of other reviewers – have greatly clarified the connection between our research questions and our

experiment. We can see that our previous manuscript could have implied that we set out to answer more questions than we did. We have tried to simplify and economize our language throughout. But to underscore and make very clear our key goal, we now state early in the introduction (p. 3) “Our first aim was to assess how these manipulations impact both overall cooperation and cooperation with participants from opposing political parties, particularly in the early stages of interaction, i.e., before network dynamics produce high levels of cooperation across conditions. Our second aim was to assess how variation in the types of reputations participants could develop - and the resulting early cooperation patterns - lead to different tendencies to form and break ties to others from one’s own versus the opposing political party.” We then outline the experimental design to explain how it addresses those questions in the Empirical Strategy section.

It does not become clear what is meant by reputations and reputation systems. It is somewhat misleading to use the word reputation, because it is not a subjective category formed by individuals about others in the experiment, but an objectively communicated image score (cf. Nowak and Milinski). As this is not clear from the beginning, the reader is a bit confused and has the false impression that the study would be about statistical discrimination, conditional cooperation, or parochial cooperation.

Very good point. We now clarify (see p. 5) how we are using the term reputation and that it is equivalent to an image score. Importantly, all research we found either uses the term image score as a “type” of reputation or, more frequently, synonymously with the term reputation. Just as one example, Wedekind and Milinski write “helping someone, or refusing to do so, has an impact on an individual’s image score within a group. This score reflects an individual’s reputation and status...” (2000, pp. 850-51) We also make clear that our operationalization is in line with the most closely related studies in the literature. Finally, we revisit this point in the Discussion section (p. 14).

It is also less clear what the expectations and hypotheses are concerning the main manipulation: the “type of reputations”. These are in fact different image scoring systems that are tested against each other. Individuals have no choice on forming reputations and the system settings apply for all subjects in a given experimental session. Therefore, it is highly misleading to talk about “a person’s treatment of ingroup and outgroup members”, “reputational information is more apt to flow between ingroup members”, and different kinds of reputations as if they were individually made. Image score systems are imposed in the experiment instead for testing the impact of image scores under different information settings (universal, parochial).

Yes, you are correct that we vary these different reputation or image scoring systems experimentally. We have revised throughout to make this clear. See, e.g., the paragraph beginning “We focus on four types of reputation systems in networks that vary by experimental condition (see Table 1)” on p. 3. We have also clarified the statement “reputational information is more apt to flow between ingroup members” to reflect the fact that we are motivating a specific experimental condition based on what often happens in the real world. It is now clear that these different reputation systems stem from our experimental manipulations, rather than the participants’ themselves.

Concerning these, as you say, “competing arguments” are tested, but it is not argued clearly on which hypotheses theoretical perspectives would agree upon and on which they would have

diverging expectations. What is the “best case” scenario? High cooperation and no political segregation? What about massive exclusions (isolation)?

A very innovative element of the theorization and of the experiment is the exploration of retaliation against outgroup members. Although the concept of intergroup conflict falls far from the design and such references are over-generalizations, this is still a very interesting question and could be justified by various streams of literature.

You are correct that we were unclear about which theoretical perspectives the competing arguments stemmed from. We now make this clear on pp. 4-5 (where we describe the implications of social identity theory vs. unbounded indirect reciprocity). We then revisit these competing arguments in the Discussion section (see pp. 11-12). We think it is now clearer that the “best case” scenario is one where political segregation is minimal and cooperation is high.

We agree that our previous mentions of intergroup conflict entailed some over-reaching. Thus, we have eliminated most references to retaliation against outgroup members, since we do not observe high levels of this, given that we intentionally selected a relatively cooperative setting (i.e., by embedded participants in dynamic networks). We now mention the possibility of retaliation against outgroup members only once (p. 4) in discussing implications of social identity theory for our setting.

Inconsistencies in the design

Papers cited by the literature review typically consider binary cooperation actions and binary reputations (good or bad) – and it would have been easier to remain at these simple settings to test fundamental theoretical hypotheses on reputation-based cooperation and political segregation. Cooperation, however, is not binary, but continuous in the experiment.

You are correct that some studies in this area use binary decisions, but some also use continuous decisions (e.g., Shirado et al. 2019; Melamed et al. 2022; Tsvetkova et al. 2022). We opted for a continuous measure since it provides more power (Gelman and Park 2008), and also allows for the emergence of more subtle behavioral differences toward ingroup vs. outgroup members. Finally, we believe a continuous measure better reflects decision-making in real world contexts where cooperation is often matter of degree. We have now clearly justified this decision (see p. 10 of the SI) and we appreciate you pushing us to do so!

The presented design considers endogenous network dynamics. Why did you need network dynamics at all to test questions on conditional cooperation?

We are interested in both conditional cooperation and network dynamics (i.e., how patterns of forming and severing ties results in political segregation). We could not study which types of reputation systems lead to more or less political segregation without allowing endogenous network dynamics. We think this is now clearer based our revisions to other points (see, in particular, our responses to your initial comments).

What you have is certainly interesting concerning network dynamics, but this comes with the loss of control in the design and complications for individual-level analysis (for some

methodological solutions, see for instance, Ule, A. (2008). Partner choice and cooperation in networks: Theory and experimental evidence (Vol. 598). Springer Science & Business Media.). Tie formation (unlike tie deletion) is regularly assumed to require mutual consent, which creates some difficulty for experimental designs. The current design is also based on mutual consent for tie formation, but with a limit of a total of four opportunities to cut and add ties. (But how did the max. 4 determine which choices are executed at the end? What was the exact algorithm, for instance if somebody received 20 requests?) A further complication for dynamic network experiments is that information is needed to make a justified choice for establishing a new tie. The details of how these difficulties are handled in the experiment are not explained. Moreover, it is not clear how tie deletion and formation phases followed each other.

Thank you for pushing us to be clearer about these issues, several of which other reviewers also asked about. We addressed the issue of mutual consent for tie formation but unilateral decisions in tie dissolution in response to Reviewer 1. We are now explicit on p. 15 of the SI that participants could potentially receive tie requests from everyone in the network that they were not already tied to and that they could accept any number of those (from 0 to all tie requests). Finally, we now explain more clearly on p. 7 of the main manuscript (see also pp. 14-15 of the SI) that, in the tie addition phase, participants were given information on the reputation or image scores of potential new alters. On the same screen, they were also given the political identities in all conditions but the control.

What is meant by “total number of points”? Total MUs earned? (This is trivial, but it is not explained in the text). For the “reputation system” manipulations, a table would help to clarify the design differences.

Yes, by total number of points we meant total number of MUs at that point in the study. We have clarified this throughout both the Main Text and the SI.

And we agree – we added Table 1 to clarify differences between conditions. Thank you very much for the suggestion!

Subjects were not presented with the full network of colored nodes (this is the impression of the reader from the text), but in fact they knew their network neighborhood only (Supp. Mat.). This is not clear from the text (especially concerning tie formation). A more precise description of what information was available would be essential to understand decisions of tie formation.

We see in retrospect that we were not clear on several key aspects of the procedure and have revised accordingly. We revised the entire Methods section in the main text. But see p. 6 for revisions we made to address this point specifically. We appreciate you pushing us to be clearer in our description of the procedures.

It seems that the network size has not been fixed between sessions. This creates a problem for the internal validity of the results. Furthermore, random graphs have the problem of differences in individual structures at the outset, hence individual decisions are no longer comparable. Why didn't you start from a regular graph?

Our paper builds on a relatively well-established research program and uses the same software as many studies before ours. We now clarify this design decision (see p. 7 of the main text) as firmly rooted in the most closely related literature (Rand et al. 2011; Melamed et al. 2018; 2020; 2022; Harrell et al. 2018; Shirado et al. 2013; Nishi et al.

2015). *This research program investigates the emergence of cooperation and segregation (in our case) in complex systems of networked humans. Other graph structures impose order that might impede or alter the evolution of the complex system. Indeed, initial network topology shapes emergence in cooperative networks, as we have shown in some of our prior work (Melamed and Simpson 2016), and now note on p. 7.*

The proportion and distribution of Democrats and Republicans in the network could also make a difference. Although this could be balanced out with a very large N, but statistical control would still be necessary to rule out effects of distortions in the (structural) distributions.

Thank you for this comment. Reviewers 1 and 3 raised similar issues. As noted in Response to Reviewer 1, we have added an extensive section to the SI on Sensitivity Analyses that addresses precisely these points. This section tests for moderation by respondent political affiliation in all our models. These analyses show that our main conclusions are robust to moderation by political identity, moderation by the more “extreme” participants (i.e., those who rate themselves at either end of the ideological continuum we used to measure political identity), and moderation by gender.

Given the strategic nature of tie deletion and tie formation, I suspect that insisting tie initiations could also been acknowledged by conditional action and tie deletions by the partner could result in some frustration and retaliation or generalized (indirect) retaliation.

With apologies, we do not understand this point. If the point is that participants might be motivated to retaliate against those who did not “approve” a tie request from them, this would not be possible since retaliation (in the limited sense of not cooperating) could only happen if a tie was formed. We did not make any changes to this since we were not certain that this is what you intended.

Limitations

There are some natural limitations of the study that need to be emphasized.

There is no major difference in the observed cooperation rates across image score system conditions (Fig. 1A). A lot of statements still emphasize differences, which are over-interpretations. Repeating the game and the availability of endogenous partner choice had a much larger impact on cooperation in every condition. There is a natural tendency of defectors becoming isolated, which drives cooperation in repeated social dilemma games with partner choice or exclusion. Furthermore, it seems that objective image scores work in general and enable a high level of cooperation in any condition.

We put a lot of effort into being clearer about study limitations in this draft (see, e.g., the extensively rewritten Discussion). That said, we do not view the fact that differences in cooperation levels between conditions disappear over time as a limitation, something that we think is now clearer. Rather, we note multiple times early on that, due to the powerful effects of network dynamics on cooperation (as you suggest), we expected differences in cooperation between conditions would primarily occur early in the study. You are correct that our language was less clear on this at other points. We have looked for multiple places to clarify this, including the revised Abstract, and pp. 3, 8, 11, 12 and in the new Table 1. We think this is now much clearer.

A multilevel model on individual decisions with appropriate controls could be part of the main text.

Agreed. We have now moved Table S1 (which predicts cooperation in rounds 1-8) to the main text. It is now Table 2.

To what extent could individuals make statistical generalizations about average group reputations and their variance? Is there a tendency for statistical discrimination?

With apologies, we do not understand the first question, in particular. We think you may be asking whether participants could sample from the Democrats and Republicans to whom they were tied to estimate “typical” cooperation levels from Democrats and Republicans. (And if they could, could they selectively form ties to these others -- i.e., do they statistically discriminate – based on those generalizations?) While this is certainly possible, we have no way of knowing the extent to which participants did this. Again, we apologize if we are misunderstanding you.

A speculation on whether Democrats (or Republicans) had a status advantage is missing. Based on status characteristics theory or based on double standards, Democrats might have an advantage of being perceived as more cooperative already at the outset. There might be double standards with retaliation as well as it could be more inappropriate for a Democrat to be a free rider than for a Republican.

This is an interesting point! We now note in the new Sensitivity Analyses section of the SI that Democrats gave more, on average, and received benefits from doing so, but we do not interpret this as a status effect, since status effects would require consensus among Democrats and Republicans beliefs about who is “better,” more competent, etc. (see, e.g., Ridgeway, Berger and other status characteristics scholars). In contrast to the consensus required for status effects, social identity effects involve each side thinking that it is “better” (more moral, etc.), as the literature we review in the paper shows is the case for political identity.

Finally, we do not find any evidence of a double standard when it comes to Democrats or Republicans. Specifically, Table S11 includes conditional logistic regressions predicting who participants dropped. Here we do not find a main effect of the participant’s political identity, nor do we find any evidence of an interaction between the participant’s political identity and experimental condition, or between participant’s political identity and whether the dropped alter shares the same political identity.

Complications arise with possible concerns of second and higher order norms. Defections might be considered right if they target an individual with low reputation (or an out-group member). The design unfortunately cannot exclude the complications arising from justified punishment and other higher order social norms.

This is a good point. But note that we think we would primarily need to be concerned about this issue had we used a design, used in some prior studies of cooperation in networks (e.g., Wang et al. 2012), where participants made a single decision each round that applied to all their alters. We think several features of our design eliminate the extent to which this would be a concern with our study. First, in our design, participants made a separate decision for each alter to whom they were tied – thus, an alter who

sought to punish their non-cooperative alters or their alters from the outgroup would not, in doing so, inadvertently “punish” ego as well. Further, participants did not get feedback after each round on how their alters treated their alters – ego was only told how their alters treated ego. Finally, during the tie deletion phase, participants had information on how their partners treated them, but not on how their partners treated their partners (alters’ alters). Thus, these higher order norms could not have been at play during these phases.

The only place where this arises is the reputation scores of potential alters. But one of our key points is that global reputation scores (i.e., those that are typically assumed in the literature on reputations) may send mixed messages, since they do not distinguish between (potentially favorable) treatment of ingroup members and (potentially less favorable) treatment of outgroup members (see our discussion of this on p. Y). Thus, we do not consider this a limitation, but a key part of our contribution. We think the revisions we have made to the Methods section and the Introduction (both detailed above in our responses to other points) makes these issues much clearer.

Reviewer #3 (Remarks to the Author):

This paper investigates how reputation systems interact with group identities in the formation of networks and cooperation. In particular, the authors study network dynamics among polarized groups (democrats vs republicans) across three experimental treatments: parochial reputations, colorblind reputations, and inter/intragroup reputations. The authors find that people are more cooperative when they can learn the reputations of both ingroup and outgroup members, and that such networks show also less segregation between political groups.

I think this is an excellent study that provides important contributions for both theory and practice. As the authors mention, they find compelling evidence that reputation systems are unbounded, a result that does not support social identity perspectives but rather recent findings that highlight the importance of reputation in increasing cooperation with outgroup members. At the same time, the authors also offer a powerful tool (reputation systems), that can be used to decrease political polarization. Moreover, I find the statistical analyses sound, and the paper well written.

Thanks for the kind comments!

I have a few suggestions that I think might enrich the paper and clarify some aspects of the methodology.

Individual heterogeneity. Previous research found that political ideology moderates parochial cooperation (Balliet, et al, 2018, Journal of Conflict Resolution; Romano et al., 2021 Phil Trans B). I was wondering whether the authors could check whether they find systematic differences between democrats and republicans in how they deal with and react to reputation systems. I understand that the authors have a majority of democrats, but I guess it would still be important to discuss such potential differences. Someone might claim that these results (e.g., the positive effect of the intra/intergroup network) are driven by the fact that the majority is democrat. Can

the authors address this potential concern with the data? If not, I would recommend to include this as a potential issue/limitation in the discussion section.

Extreme cases. Relatedly, how are people in the extreme of the political ideology scale behaving compared to more moderate people? I guess that to defend social identity approaches, one might argue that the inter/intragroup finding is driven by people that responded 3 or 4 (hence the middle points) in the initial political ideology question. Also in this case, I would suggest the authors to check whether they can address this potential issue with additional data analyses, or mention this as a potential limitation in the discussion section.

Thank you for both of these suggestions. Reviewers 1 and 2 raised similar questions about moderation. As described more fully in our earlier responses, we added a new section “Sensitivity Analyses” to our SI where we assess whether any of our results are moderated by political identity (being Democrat or Republican), political extremes, or participant gender. We do find some of the effects that you anticipate - e.g., amplified homophily effects among extremes – in some analyses. But once appropriate controls are included, these factors do not moderate our results in any substantial way. We appreciate you encouraging us to look into these issues, as it yields more confidence in the conclusions in our Main Text.

Methodology/design/sample. I think that the authors could provide more details about the data-collection and methodology. I find remarkable that the authors were able to collect such amount of data for real time interactions across 18 rounds. With such large sessions, I find difficult to believe that the authors did not encounter drop outs in their data collection. Maybe I missed this, but I would suggest to report any drop out rate encountered during the sessions, and discuss how this might have potentially affected the results.

Thank you for raising this concern. Indeed, it was an oversight to not discuss drop-outs. We now describe them in a new Missing Data section in the SI. Drop-outs could have occurred for several different reasons: participants might have voluntarily left the study, participants’ internet connection dropped, or they took too long to make a decision. In short, they were uniformly distributed by experimental condition and political affiliation. (And to your more general point, we have also added much more detail about our methodology, design, and sample throughout, also in response to comments from the other reviewers.)

Reviewer #4 (Remarks to the Author):

The goal of this paper is to empirically test the effect of identity-based reputations – here political identity – on network formation and cooperative behavior within these networks. The empirical analysis seems to be very well done and the conclusions are clear: allowing users to develop reputations in terms of how they treat out group members leads to more cooperative behavior and more inclusion of outgroup members within networks than reputations that are either based only on how one treats ingroup members or are “colorblind”, in the sense that the reputations do not distinguish between how users treat outgroup members from ingroup members.

My sense is that the research is interesting and well executed, and certainly speaks to a very important problem in contemporary US society – the tendency of political polarization to drive people into networks that are increasingly segregated by ideology, with the assumption that this will in turn lead to further polarization. I think it therefore is certainly worth consideration by the editors for publication in Nature Communication, but I want to be very clear that I am not a network scientist, so I can not speak to the quality or the appropriateness of the network methods employed by the authors. From that perspective, though, I would like to offer three suggestions that I think would help make the paper stronger – and have more of a contribution to a wider audience – in a revised version of the manuscript.

Thank you for this summary and the generous comments about our research.

First, the paper is at the same time both dense and a bit repetitive. I had to read the manuscript several times before I understood the distinction between the three different reputation mechanisms and the various tests of the hypotheses. I think there is a way to do all of this in a much more streamlined way, perhaps involving a table that shows the relationship between the different reputation mechanisms and the various dependent variables, showing exactly what is predicted in each case.

Adding a table is a great suggestion, one also made by Reviewer 2. We have now done so. We have also looked for opportunities throughout to make the paper less dense and repetitive. We think the paper is now easier to follow, while also being less repetitive.

Relatedly, were these hypotheses pre-registered? Apologies if I missed it, but I could not find anything in the text about pre-registration.

We did not preregister our hypotheses. While we recognize disciplinary norms are not valid excuses, it is not yet the norm in our discipline (sociology) to pre-register hypotheses. We based our hypotheses off direct extensions of prior work, especially by Romano et al (2017a; 2017b). That these predictions are rooted in prior work is now clearer (see pp. 4-5).

Second, and perhaps most importantly, I think the paper would benefit from a deeper discussion of the external validity of findings related to a game where there are incredibly clear benefits from cooperation. As the authors note, everyone figures this out eventually and cooperation goes up over rounds. So the question is, what does this capture about the motivating example of the paper: the seemingly unstoppable rise of political polarization in the United States? What is the game everyone is playing where there are clear benefits from cooperation? If anything, from a political science perspective, we tend to assume that political competitions are zero-sum: if my side gets control of the government and implements my preferred policy outcome, then by definition the other side does not get control of the government and can not implement their preferred policy outcome. I am willing to accept the results of the experiments as confirming the authors' hypotheses, but I think the potential impact of the paper would be much larger if the authors could make a compelling argument about what system they are actually capturing with their experiment. In other words, are we learning anything outside the confines of the experimental treatments?

These are very important questions and we have almost completely rewritten the Discussion, especially the latter half, to tackle them head on. Rather than describing all our changes and additions here in this revision memo, we'll point to one example here.

We now clarify that in highly polarized contexts, people tend to think about the outcomes of interactions over ideological issues in zero-sum ways, exactly as you suggest. We now cite relevant work on this topic. But we then note that political ideology also plays a role in “cold” economic calculations (i.e., in the types of decisions that our participants are making), again citing relevant work on this topic. We then describe how our findings build on this work. (See p. 13.) Relatedly, at another point in the Discussion, we discuss the upshot of this finding when we link our finding that intra/intergroup reputation systems generate more ties to outgroup members with a key message from theories and research on intergroup contact theory – namely that ties to outgroup members occurring in more cooperative contexts (like the one we study) may end up reducing distrust and animosity toward the outgroup as a whole, thus potentially reducing the sort of zero-sum thinking that we’d otherwise expect in polarized environments. (See pp. 12-13.)

We think the revised Discussion ends up doing two key things much better than the previous one. First, it better clarifies what our findings may say outside the context of our specific study. But it also more explicitly recognizes where we are less certain about the external validity of findings from our study. In the latter case, we explicitly point to specific directions for future research that could address whether and when our results would apply to different settings and when we would expect something different. In our view, there are a lot of interesting possibilities here. Indeed, we are now much more excited about a program of research building on this paper than we were when we developed the original draft of the paper! We thus appreciate you pushing us to engage more deeply with these types of questions.

Third – and related to the previous point -- what are real world institutions that could privilege these types of reputations? This is where the paper ends up: arguing for institutions that privilege reputation building that distinguish between how one treats in groups and out groups, and against reputations that are color blind or focused on how one treats only one’s own group. Fair enough, but what are examples of such institutions, including both institutions to be avoided and institutions to be recommended? It does seem to me that there is an interesting opportunity here for the authors to discuss the differences between offline and online networks. Off the top of my head, my sense would be that these types of institutions would be easier to design in online networks – where platforms can tinker with the rules of the game – than they are in offline networks, but either way I think it would behoove the authors to offer their thoughts in this regard.

This is a great question and we appreciate you raising it. We fully agree that such a system would be easier to implement in online settings, an issue that we now address in the revised Discussion (see p. 14). We note that one possible way social media platforms (for instance) could implement our intra/intergroup reputation system would be the implementation of “civility scores” that denote positive treatment of outgroup members. We note in the discussion that the implementation of civility scores by social media platforms could signal the importance of positive treatment of outgroup members through such reputations, providing a “top down” signal that prosocial behavior toward outgroup members is normative and expected (in contrast to the norm of cross-party antagonism that has emerged in existing social media platforms). These specific predictions could, of course, be tested experimentally.

Minor point: I would recommend labeling the units on the Y-Axis in the figures. For example, in Figure 1 make it clear that these are the monetary units (at first glances, it looks like it could be percentages).

Thanks for making this suggestion. We have added units for clarity.

Reviewers' Comments:

Reviewer #1:

Remarks to the Author:

This manuscript is much improved and ready for publication. The authors are to be commended for the thoroughness of their revisions and response. Below, I suggest a few optional cosmetic fixes for matters that arose during the course of their revisions:

- In Table 1, "Politics Salient" seems a bit cryptic so I would change it to "Political Identity" and replace "yes" and "no" with "visible" and "hidden" (or do something pretty similar). Also, I would break the theoretical expectations column into two---one for cooperation and once for network--- unless there's no reasonable way to do this within the page width.
- This may seem like a personal preference, but I'd suggest replacing "salient" with "visible" throughout the text as well. In the political psychology literature at least, "salient" usually implies that everyone has access to a piece of information but the treatment group is reminded of it (or has it highlighted for them). If the authors prefer to keep the word salient the manuscript is still readable, but I do think this change will improve clarity.
- In the descriptions of the parochial condition, I started to lose track of what "ingroup" meant: ingroup from the point of view of the ego (i.e. ingroup = MY group) or ingroup from the point of view of the prospective partner (i.e. ingroup = THEIR group)? I'm pretty sure the correct answer is THEIR group. An example would make this crystal clear and easy to remember: "Players in the parochial condition saw how the Democrats had treated their fellow Democrats and how the Republicans had treated their fellow Republicans but not how they had treated members of the opposite party." Or something like that. This could go in the intro or methods section.
- Authors accidentally wrote "points" to "MUs" in two of the sentences they added. Just do a find-and-replace to catch them all.
- The clarity of the Methods section is much improved. The one paragraph that I'm still having to read multiple times in order to comprehend is the description of what the players see during the tie addition phase. Here is some language I suggest. Feel free to modify or use it verbatim:

"... and the list of prospective alters. In all four conditions, participants could see prospective alters' total number of MUs. In the three experimental conditions, they could also see their political affiliation and the average number of MUs they gave their neighbors—either all neighbors, only ingroup neighbors, or ingroup and outgroup neighbors tallied separately—in the previous three rounds. These averages represent the alters' reputations (i.e. their level of cooperation in previous rounds). Hence, which version participants saw depended on the experimental condition to which they were assigned (see table 1)."
- In the sentence that reads "there were no differences in clustering between conditions at the end of the study", I'd say "level of clustering" perhaps with "level" in italics and then "basis" in italics in the next sentence. Again, not crucial, but improves readability.

Reviewer #2:

Remarks to the Author:

Thank you very much for the careful response to my previous comments and suggestions. Many of your responses are satisfactory, but there are still some issues that prevent publication of the manuscript in its current shape. There are requests from the first round where I was less satisfied with your response:

- I agree with Reviewer #1 that homophily should be seen as a cause and not as an outcome. The formulations should be even clearer.
- Likewise, I share Reviewer #1's concern that regression in a network setting is problematic. I suggest that you follow Reviewer #1's advice more closely.
- You had problems of understanding this comment of mine: "Given the strategic nature of tie deletion and tie formation, I suspect that insisting tie initiations could also be acknowledged by conditional action and tie deletions by the partner could result in some frustration and retaliation or generalized (indirect) retaliation." Apologies if I was not clear. What I meant is as follows. Tie deletion can be handled easily because if one of the parties delete a tie, then the tie ceases to exist. Tie formation, however, should be approved by the other party. When a tie initiation is not approved, it could be a form of retaliation by the receiver, and it could also be remembered by the sender and be later a reason for not cooperating. This possibility is not handled at all by your analysis. It is also not clear if the memory of the participants is facilitated with supporting information (e.g., history of denied tie formation requests) on their screen in the experiment. Furthermore, the fact that every proposal had to be confirmed by the prospective alter implies the presence of strategic considerations for participants about their future ideal network composition. This should at least be acknowledged in the introduction, potentially making a reference to games of network formation (e.g., Jackson 2008) and particularly to those that connect cooperation and endogenous network dynamics (e.g., Takács et al. 2008).
- I still miss an explanation from the main text why you needed a continuous game.
- Thank you for making a direct reference in the text that this type of reputation is sometimes referred to an "image score".
- As I indicated in my last comment in Round 1, complications arise with possible concerns of second and higher order norms. You could make a reference to higher order norms and why they were disregarded (e.g., for the sake of simplicity).

Further comments:

- You might want to highlight that there were repeated cooperation experiments before this one in which participants could build and sever ties (such as Ule 2008; or Takács & Janky 2007; Goeree et al. 2009; Bravo et al. 2012).
- It should be more precise what is meant by network clustering and segregation. For unambiguous definitions, see for instance Mepham (2022).
- Multiple observations are embedded within the individual (31,442 observations are embedded in 1,073 participants). While you have separate variance elements at the decision (NOT the participant!) and at the network level, the proper multilevel (hierarchical) regression should consider the individual level above the decision level first.
- For analyzing dynamic changes in the network, you could use a model that explains new tie formation / maintenance / deletion in time t with the t-1 situation. Models in Tables S1 and S2 give a good first impression, but they do not control for network dependencies or at least for the network ties present in t.

Minor comments:

- Abstract: Place (N=1,073) after participants and not after networks.
- It should be noted at the first mention of "prisoners' dilemma" in the Introduction that the paper is concerned about dyadic symmetric trust games. The emphasis should be repeatedly on the dyadic nature of cooperation interactions, as this is a very important design feature.

References

- Bravo, G., Squazzoni, F., & Boero, R. (2012). Trust and partner selection in social networks: An experimentally grounded model. *Social Networks*, 34(4), 481-492.
- Cuesta, J. A., Gracia-Lázaro, C., Ferrer, A., Moreno, Y., & Sánchez, A. (2015). Reputation drives cooperative behaviour and network formation in human groups. *Scientific reports*, 5(1), 7843.
- Goeree, J. K., Riedl, A., & Ule, A. (2009). In search of stars: Network formation among heterogeneous agents. *Games and Economic Behavior*, 67(2), 445-466.
- Jackson, M. O. (2008). *Social and economic networks* (Vol. 3). Princeton: Princeton University Press.

Mephram, K. (2022). Political Opinions and Offline Social Networks in a Swiss Student Community (Doctoral dissertation, ETH Zurich).

Takács K. & Janky B. (2007). Smiling Contributions: Social Control in a Public Goods Game with Network Decline. *Physica A*, 378 (1): 76-82.

Takács K.; Janky B., and Flache, A. 2008. Collective Action and Network Change. *Social Networks*, 30(3): 177-189. <https://doi.org/10.1016/j.socnet.2008.02.003>

Ule, A. (2008). Partner choice and cooperation in networks: Theory and experimental evidence (Vol. 598). Springer Science & Business Media.

Reviewer #3:

Remarks to the Author:

The authors addressed very well all the comments raised in the first round of the review. I think the paper provides an important contribution to both theory and practice. I have no further comments.

Reviewer #4:

Remarks to the Author:

I want to commend the authors for addressing my previous comments so thoroughly. The paper is *much* easier to read now – just much, much clearer. As such, it is much easier for me as a political scientist (and not a network scientist) to understand what exactly is happening here and what is being claimed. The authors have also addressed my requests regarding a discussion of external validity and real-world applications by thoroughly rewriting the discussion section. As such, I would consider my original requests to have been addressed.

I have a few more comments, though, that I think the authors should address in a final round of revisions.

First, regarding the discussion section, the authors – drawing on both my requests and points raised by other reviewers – have done an admirable job of attempting to identify real-world applications of their key finding (i.e., segregation and perhaps polarization could be reduced by systems that provide reputations based on the treatment of outgroups) along with caveats about applying the results of these experiments in those real-world cases. However, I must admit, I find myself not convinced by the arguments that what is learned from these games with monetary rewards from cooperation will apply on social networks. To put another way, I think the authors now do such a good job laying out the caveats in this regard highlighted by other reviewers, I'm just not sure that there is a there there now. The goal of this game was to earn money, so learning that someone treats out-group members better has a direct financial payoff if you are an out-group member deciding whether to sever a tie. While a civility score might capture that idea, it does not capture the idea that the attention economy (which does not have a fixed budget constraint – I can like a post by someone else without diminishing my capacity for future likes) could also feature in-groups that compete for *low* out group civility scores, especially among political elites. I mean, couldn't you imagine a politician boasting about a low out-group civility rating the same way they boast about a high or low rating from the NRA? To be clear, I am not saying this is not a reason to not publish the article, nor do I think that the authors are somehow obscuring this point. I just am registering that I am not sure they have thought through just how difficult it might be for a system like this to produce the desired outcomes outside of a game where are purely monetary and based on cooperation.

Second, I now have a bit of a concern with the way the results are described vis a vis statistical significance (which again is a credit to how much easier it is to follow the revised version of the paper!). A few points:

- The claim in line 260-62 that the effect was only statistically significant for parochial treatment is true if you are using $P < .05$, but the other two effects are pretty close to that cut off, especially for the colorblind treatment. Also, my guess is that that the difference between some of these effects may not themselves be statistically significant. So I'd just rely on the Figure which shows a much larger and statistically significant effect for parochial and leave it at that.

- Line 274: Similarly saying "there is no statistically significant effect" when $p < .10$ and is in the direction of the effect that you are claiming does not exist might be pushing too hard on asterisks.

- Line 278 – Saying those in the color blind condition were "marginally more likely to do so" does not really seemed justified at all from Figure 2a – those distributions look basically the same.

So in general, I'd dial back these claims about differences to be a little clearer that you are looking at general effect sizes relative to each other and not fixating on whether something is slightly above $p = .05$.

But I do have another question related to Figure 1b – why are the confidence intervals so much smaller than in the first submission of the paper? Was there an original coding error? Just trying to make sure there's not an error now in the new Figure. Additionally, it is difficult to read Figure 1b now – the difference between the two colors is not easy to pick up. If you have to go monochrome, can you use different shapes and make them bigger? And fwiw, I'd use the same scale for the y-axis on Figures 1a and 1b – would make comparisons across the two easier.

A couple other minor points:

Line 186 – this is not how we typically think of prisoner's dilemma game in political science. PD games involve a choice of defection or not – they do not give opportunities to share varying amounts of money. I think what you've got here is actually what political scientists called "dictator games" where one player distributes money to a bunch of other players and keeps what is left over, although I personally have never seen a dictator game where everyone is simultaneously a dictator. I'd call this a "networked dictator game", but I'm not a game theorist so maybe someone else has already come up with a name for it? But calling it a PD is going to confuse some readers.

Line 205-7: So how often do people lose all their ties? And how might this impact inference? Apologies if you have described this somewhere else, but I think this is important to share with the reader.

Closely related to the previous point, why are there so few Republicans in Figure 4b "parochial network" (lower right-hand figure)? Is that b/c they lost all their ties? This seems worth addressing.

Revision Memo

“Reputation, Cooperation, and the Emergence of Political Segregation in Networks”

Once again, we appreciate the opportunity to revise and resubmit our manuscript to *Nature Communications*. The reviewers’ comments and suggestions were again very helpful and, as detailed below, their suggestions led to key improvements. Below, we reproduce the Reviewer comments verbatim and respond to them in indented, *italicized*, text.

Following Nature Communications policy, we have submitted both a “clean” version of the revised document and a “tracked changes” version that highlights all changes since the previous submission. So that it is clear where we made revisions, the page numbers below refer to the manuscript with tracked changes.

In addition to the changes made in response to the reviewers’ feedback, we also reformatted the paper to be in line with *Nature Communications* manuscripts. The biggest change in this regard was moving the Methods section to the end of the paper (after the Discussion), per journal policy (along with some additional edits for clarification and streamlining that resulted from this restructuring).

REVIEWER COMMENTS

Reviewer #1 (Remarks to the Author):

This manuscript is much improved and ready for publication. The authors are to be commended for the thoroughness of their revisions and response. Below, I suggest a few optional cosmetic fixes for matters that arose during the course of their revisions:

- In Table 1, “Politics Salient” seems a bit cryptic so I would change it to “Political Identity” and replace “yes” and “no” with “visible” and “hidden” (or do something pretty similar). Also, I would break the theoretical expectations column into two---one for cooperation and once for network---unless there’s no reasonable way to do this within the page width.

We liked both these suggestions and implemented the language change. We tried to break the theoretical expectations into two columns. But, as the Reviewer anticipated, the table ended up being too wide for the page. See revised Table 1.

- This may seem like a personal preference, but I’d suggest replacing “salient” with “visible” throughout the text as well. In the political psychology literature at least, “salient” usually implies that everyone has access to a piece of information but the treatment group is reminded of it (or has it highlighted for them). If the authors prefer to keep the word salient the manuscript is still readable, but I do think this change will improve clarity.

This is also a good suggestion. We have changed salient to visible in our methods and results sections, as well as in the SI. At points in the introduction and discussion,

“visible” felt a bit awkward, but we agree that “salient” also confused matters. Thus, we used other intuitive labels – e.g. “known” - in the Introduction (e.g., p. 3) and Discussion (p. 13).

- In the descriptions of the parochial condition, I started to lose track of what “ingroup” meant: ingroup from the point of view of the ego (i.e. ingroup = MY group) or ingroup from the point of view of the prospective partner (i.e. ingroup = THEIR group)? I’m pretty sure the correct answer is THEIR group. An example would make this crystal clear and easy to remember: “Players in the parochial condition saw how the Democrats had treated their fellow Democrats and how the Republicans had treated their fellow Republicans but not how they had treated members of the opposite party.” Or something like that. This could go in the intro or methods section.

That is the correct answer, and an example is a good idea. We have added one in the Methods section (p. 14).

- Authors accidentally wrote “points” to “MUs” in two of the sentences they added. Just do a find-and-replace to catch them all.

Good catch. We have replaced “points” with “MUs” throughout the main text and SI.

- The clarity of the Methods section is much improved. The one paragraph that I'm still having to read multiple times in order to comprehend is the description of what the players see during the tie addition phase. Here is some language I suggest. Feel free to modify or use it verbatim:

"... and the list of prospective alters. In all four conditions, participants could see prospective alters' total number of MUs. In the three experimental conditions, they could also see their political affiliation and the average number of MUs they gave their neighbors—either all neighbors, only ingroup neighbors, or ingroup and outgroup neighbors tallied separately—in the previous three rounds. These averages represent the alters' reputations (i.e. their level of cooperation in previous rounds). Hence, which version participants saw depended on the experimental condition to which they were assigned (see table 1)."

This text does a very nice job of summarizing what participants saw during the tie addition phase of the study. We added the text (almost) verbatim in the Methods section of the main text (pp. 13-14).

- In the sentence that reads "there were no differences in clustering between conditions at the end of the study", I'd say “level of clustering” perhaps with "level" in italics and then "basis" in italics in the next sentence. Again, not crucial, but improves readability.

Again, good suggestion. We made these changes (p. 8).

Reviewer #2 (Remarks to the Author):

Thank you very much for the careful response to my previous comments and suggestions. Many of your responses are satisfactory, but there are still some issues that prevent publication of the manuscript in its current shape. There are requests from the first round where I was less satisfied with your response:

- I agree with Reviewer #1 that homophily should be seen as a cause and not as an outcome. The formulations should be even clearer.

We see now that there was still some ambiguity on this in the SI, which we have now cleared up (see, e.g., pp. 201 of the SI). We appreciate you pushing up to address all instances where we were referring to homophily as an outcome, rather than a cause.

- Likewise, I share Reviewer #1's concern that regression in a network setting is problematic. I suggest that you follow Reviewer #1's advice more closely.

We agree and altered our estimators based on Reviewer 1's comments in the previous submission. But we cannot think of a way of following that advice more closely but would be happy to consider any additional suggestions in this regard.

- You had problems of understanding this comment of mine: "Given the strategic nature of tie deletion and tie formation, I suspect that insisting tie initiations could also been acknowledged by conditional action and tie deletions by the partner could result in some frustration and retaliation or generalized (indirect) retaliation." Apologies if I was not clear. What I meant is as follows. Tie deletion can be handled easily because if one of the parties delete a tie, then the tie ceases to exist. Tie formation, however, should be approved by the other party. When a tie initiation is not approved, it could be a form of retaliation by the receiver, and it could also be remembered by the sender and be later a reason for not cooperating. This possibility is not handled at all by your analysis. It is also not clear if the memory of the participants is facilitated with supporting information (e.g., history of denied tie formation requests) on their screen in the experiment. Furthermore, the fact that every proposal had to be confirmed by the prospective alter implies the presence of strategic considerations for participants about their future ideal network composition. This should at least be acknowledged in the introduction, potentially making a reference to games of network formation (e.g., Jackson 2008) and particularly to those that connect cooperation and endogenous network dynamics (e.g., Takács et al. 2008).

Ahh, now we understand. Thank you, and great point! We addressed these comments in several ways. On p. 13, we now explain "Since ties between participants represented the only opportunities for interactions in the study, following prior work in this literature^{7,8,48,66}, those whose tie initiations were denied did not have an opportunity to retaliate against the participant who denied the tie invitation." We also note on p. 14 of the SI "Participants were not shown a history of rejected tie requests, i.e., participants were not consistently reminded of which prospective alters previously rejected tie requests. (The provided screenshots show exactly what information participants had during the tie formation phases.)" Finally, we now acknowledge that our design (specifically the dynamics portion of our design) rules out the types of strategic considerations that prior work (especially Takács et al. 2008) have addressed (see p. 14).

We appreciate you pushing us to address these points, as they help clarify our design and better situate it in the broader literature.

- I still miss an explanation from the main text why you needed a continuous game.

The reviewer is correct that while we previously included this justification in the SI, we did not include it in the main text, which would have been more appropriate. We now do so on p. 13.

- Thank you for making a direct reference in the text that this type of reputation is sometimes referred to an “image score”.

Yes, thank you for suggesting it.

- As I indicated in my last comment in Round 1, complications arise with possible concerns of second and higher order norms. You could make a reference to higher order norms and why they were disregarded (e.g., for the sake of simplicity).

Good suggestion. We now (see p. 5) explain what “standing” strategies would look like in our context and note that, following the research we build on, for simplicity, we use image scores instead, thus ruling out higher order reputational norms.

Further comments:

- You might want to highlight that there were repeated cooperation experiments before this one in which participants could build and sever ties (such as Ule 2008; or Takács & Janky 2007; Goeree et al. 2009; Bravo et al. 2012).

We now cite these papers in the first sentence of our Methods section (see p. 13).

- It should be more precise what is meant by network clustering and segregation. For unambiguous definitions, see for instance Mephram (2022).

Thank you for encouraging us to be more precise about these key network measures. We now define these terms on p. 8 of the main text, and provide references to the specific R functions we use to compute them in the “Network Measures” section of the SI (pp. 48-49).

- Multiple observations are embedded within the individual (31,442 observations are embedded in 1,073 participants). While you have separate variance elements at the decision (NOT the participant!) and at the network level, the proper multilevel (hierarchical) regression should consider the individual level above the decision level first.

Good catch. We have added the additional variance component to these models. As a result, the permutation tests about the fixed effects were recomputed, as were the bootstrapped distributions around the margins in Figure 1B.

- For analyzing dynamic changes in the network, you could use a model that explains new tie formation / maintenance / deletion in time t with the $t-1$ situation. Models in Tables S1 and S2 give a good first impression, but they do not control for network dependencies or at least for the network ties present in t .

Thank you for this thoughtful comment. In Table S1 we predict the binary decision of whether to drop an alter given the opportunity to do so. Here we incorporate network structure via the number of current partners, their average endowment, and how much they gave the participant. As we noted in response to Reviewer 1 in the previous round, we are unaware of any network models that are suitable to our data structure. In such cases, Snijders (2011) recommends using non-parametric inference or controlling for structure through covariates. In this case, we are doing both these things: inference is permutation-based and we are controlling for three network variables. If there are other network covariates that would pose a threat to validity, please let us know which ones to consider, and we will control for them too.

Minor comments:

- Abstract: Place (N=1,073) after participants and not after networks.

Good catch. Corrected.

- It should be noted at the first mention of “prisoners’ dilemma” in the Introduction that the paper is concerned about dyadic symmetric trust games. The emphasis should be repeatedly on the dyadic nature of cooperation interactions, as this is a very important design feature.

There are several important points here.

First, we fully agree that emphasizing the dyadic nature of the cooperative interactions is critical, so we now do that right way, including in the abstract and on p. 2 of the main text (this is in addition where we already emphasized this in the prior submission, later in the paper and in the SI). We think this is now much clearer early on.

Second, we had not previously come across the phrase “dyadic symmetric trust game” and could not find any reference to it (or, more simply “symmetric trust game”) in the literature. But given that Reviewer 4 also had a comment about the incentive structure (questioning whether it was a Prisoners’ Dilemma), we thought that it was important to be explicit about precisely how the incentives of interactions in our study fully satisfy the definition of a Prisoners’ Dilemma. We therefore added Table 2, which gives a payoff matrix for decisions in our study and notes that this payoff structure fully satisfies both inequalities required for a decision scenario to be a Prisoners’ Dilemma— namely $T > R > P > S$ and $2R > T + S$. (We refer readers to this table on p. 6). We appreciate the reviewer encouraging us to address these issues.

*We should add that our use of the Prisoners' Dilemma label is fully consistent with work in this literature, which we now make clearer. Further, the choices that our participants faced – deciding how much of a continuous endowment to give to an alter (who is simultaneously making the same decision) and having any given amount be doubled – is fully consistent with how sociologists and social psychologists have operationalized the PD outside of network contexts (see, as one example, Yamagishi and Kiyonari, "The Group as a Container of Generalized Reciprocity" *Social Psychology Quarterly*). Thus, our use is consistent with the literature, both in a narrow sense (studies of cooperation in networks) and broader sense (the wider social dilemmas literature).*

References

- Bravo, G., Squazzoni, F., & Boero, R. (2012). Trust and partner selection in social networks: An experimentally grounded model. *Social Networks*, 34(4), 481-492.
- Cuesta, J. A., Gracia-Lázaro, C., Ferrer, A., Moreno, Y., & Sánchez, A. (2015). Reputation drives cooperative behaviour and network formation in human groups. *Scientific reports*, 5(1), 7843.
- Goeree, J. K., Riedl, A., & Ule, A. (2009). In search of stars: Network formation among heterogeneous agents. *Games and Economic Behavior*, 67(2), 445-466.
- Jackson, M. O. (2008). *Social and economic networks* (Vol. 3). Princeton: Princeton University Press.
- Mepham, K. (2022). *Political Opinions and Offline Social Networks in a Swiss Student Community* (Doctoral dissertation, ETH Zurich).
- Takács K. & Janky B. (2007). Smiling Contributions: Social Control in a Public Goods Game with Network Decline. *Physica A*, 378 (1): 76-82.
- Takács K.; Janky B., and Flache, A. 2008. Collective Action and Network Change. *Social Networks*, 30(3): 177-189. <https://doi.org/10.1016/j.socnet.2008.02.003>
- Ule, A. (2008). *Partner choice and cooperation in networks: Theory and experimental evidence* (Vol. 598). Springer Science & Business Media.

Reviewer #3 (Remarks to the Author):

The authors addressed very well all the comments raised in the first round of the review. I think the paper provides an important contribution to both theory and practice. I have no further comments.

Reviewer #4 (Remarks to the Author):

I want to commend the authors for addressing my previous comments so thoroughly. The paper is **much** easier to read now – just much, much clearer. As such, it is much easier for me as a political scientist (and not a network scientist) to understand what exactly is happening here and what is being claimed. The authors have also addressed my requests regarding a discussion of external validity and real-world applications by thoroughly rewriting the discussion section. As

such, I would consider my original requests to have been addressed.

I have a few more comments, though, that I think the authors should address in a final round of revisions.

First, regarding the discussion section, the authors – drawing on both my requests and points raised by other reviewers – have done an admirable job of attempting to identify real-world applications of their key finding (i.e., segregation and perhaps polarization could be reduced by systems that provide reputations based on the treatment of outgroups) along with caveats about applying the results of these experiments in those real-world cases. However, I must admit, I find myself not convinced by the arguments that what is learned from these games with monetary rewards from cooperation will apply on social networks. To put another way, I think the authors now do such a good job laying out the caveats in this regard highlighted by other reviewers, I'm just not sure that there is a there there now. The goal of this game was to earn money, so learning that someone treats out-group members better has a direct financial payoff if you are an out-group member deciding whether to sever a tie. While a civility score might capture that idea, it does not capture the idea that the attention economy (which does not really have a fixed budget constraint – I can like a post by someone else without diminishing my capacity for future likes) could also feature in-groups that compete for *low* out group civility scores, especially among political elites. I mean, couldn't you imagine a politician boasting about a low out-group civility rating the same way they boast about a high or low rating from the NRA? To be clear, I am not saying this is not a reason to not publish the article, nor do I think that the authors are somehow obscuring this point. I just am registering that I am not sure they have thoughts through just how difficult it might be for a system like this to produce the desired outcomes outside of a game where are purely monetary and based on cooperation.

This is an important point and we thought carefully about various ways of addressing it. This resulted in several key changes to our Discussion. First, we cut the paragraph on civility scores, since it was particularly misleading. Part of the problem in the prior submission was that we wanted to communicate how our findings might be used to reduce polarization based on group identities in general (not specific to political identities in hyperpolarized contexts) and that was the context for our “civility score” discussion. Of course, we see now that any discussion of civility scores – particularly online – inevitably suggests that we think our findings could be straightforwardly extended to the very difficult context of reducing partisan animosity on social media. That would clearly be a big stretch, as the reviewer correctly notes.

Second, we have tried to more clearly emphasize that whether our findings generalize as we move further away from contexts where there are at least some common goals (as theorized by intergroup context theory, as we discuss on pp. 10-11) to the (perceived) zero sum contexts of contemporary American politics is very much an open question. We are much more cautious about extrapolating to these more difficult cases.

As a result, of these first two changes, our Discussion now more clearly distinguishes two types or levels of implications: i) more cautious implications for applying these findings to political identities, and ii) a more basic research understanding of how reputation systems work when they occur in the context of social identities in general (rather than highly polarized political identities that we see in the U.S.).

Finally, and perhaps most importantly, we now take much more seriously just how difficult reducing political polarization and animosity is. To do so, we have brought in a discussion of the recent review paper by Hartman et al. (2020, "Interventions to Reduce Partisan Animosity, Nature Human Behaviour). We note, as [Hartman et al.] find, "any single intervention is unlikely to be effective by itself, given that partisan animosity arises from, and is reinforced by, processes at three different levels – thoughts (e.g., misperceptions of the other side), relationships (relative absence of ties across party lines) and institutions (norms and political structures that promote partisanship). Thus, beyond the question of whether our findings would apply in more acrimonious, zero-sum, contexts, our work only addresses how different types of reputation systems impact relationships across party lines. It does not address thoughts or institutions. And while a central tenet of intergroup contact theory is that ties to outgroups can reduce outgroup prejudice and misperceptions (i.e., change thoughts), the arguably more difficult problem of changing institutions would remain." See p. 11. We also anticipate this difficulty early in the paper (p. 3)

The revised Discussion thus provides a more honest assessment of how our findings may be applied, while explicitly recognizing the need for a multi-prong approach to reducing polarization and animosity. Further, we think the revised discussion is now clearer that the importance of our contribution does not hinge on the ability to directly translate our findings to reducing political polarization in the real world. The more general basic research implications that result from bridging reputation processes with intergroup and network processes remain and, we believe, are important.

Second, I now have a bit of a concern with the way the results are described vis a vis statistical significance (which again is a credit to how much easier it is to follow the revised version of the paper!). A few points:

- The claim in line 260-62 that the effect was only statistically significant for parochial treatment is true if you are using $P < .05$, but the other two effects are pretty close to that cut off, especially for the colorblind treatment. Also, my guess is that that the difference between some of these effects may not themselves be statistically significant. So I'd just rely on the Figure which shows a much larger and statistically significant effect for parochial and leave it at that.

Thank you for this comment. We agree that the paragraph in question focused too heavily on the arbitrary $p < .05$ threshold. We revised our language to discuss the relative strength of the effect, as illustrated in the Figure. (See p.7)

- Line 274: Similarly saying “there is no statistically significant effect” when $p < .10$ and is in the direction of the effect that you are claiming does not exist might be pushing too hard on asterisks.

- Line 278 – Saying those in the color blind condition were “marginally more likely to do so” does not really seem justified at all from Figure 2a – those distributions look basically the same.

So in general, I’d dial back these claims about differences to be a little clearer that you are looking at general effect sizes relative to each other and not fixating on whether something is slightly above $p = .05$.

We fully agree with these points and revised our summary of results accordingly (see, p. 8).

But I do have another question related to Figure 1b – why are the confidence intervals so much smaller than in the first submission of the paper? Was there an original coding error? Just trying to make sure there’s not an error now in the new Figure. Additionally, it is difficult to read Figure 1b now – the difference between the two colors is not easy to pick up. If you have to go monochrome, can you use different shapes and make them bigger? And fwiw, I’d use the same scale for the y-axis on Figures 1a and 1b – would make comparisons across the two easier.

In the original submission we relied on parametric inference. The original Fig 1B was generated using delta method standard errors. Based on comments from Reviewer 1 in the previous round, we transitioned to nonparametric inference. The revised Fig 1B is generated using a bootstrapped confidence interval. These are indeed different estimates. We have also updated the colors and scales in Fig 1 more generally.

A couple other minor points:

Line 186 – this is not how we typically think of prisoner’s dilemma game in political science. PD games involve a choice of defection or not – they do not give opportunities to share varying amounts of money. I think what you’ve got here is actually what political scientists called “dictator games” where one player distributes money to a bunch of other players and keeps what is left over, although I personally have never seen a dictator game where everyone is simultaneously a dictator. I’d call this a “networked dictator game”, but I’m not a game theorist so maybe someone else has already come up with a name for it? But calling it a PD is going to confuse some readers.

Reviewer 2 raised a similar concern, and we see how our previous drafts were confusing the issue. Generally speaking, any money given to others in a dictator game is not subject to a multiplier. In our study, any amount given is doubled before alter receives it. It is precisely this multiplier -- and the fact that alter is making the same decision with respect to ego simultaneously - that makes the incentive structure of our study PD. (Indeed, as noted in our response to Reviewer 2, there is a long tradition in sociology and social psychology of operationalizing PD just as we do here. Further, all studies in the networks and cooperation literature that we are aware of refer to this incentive structure as PD.) That said, we see that this was not at all obvious in the previous draft. We therefore introduced Table 2, which shows exactly how the incentive structure faced by participants in our study conforms to the two defining inequalities of a Prisoners' Dilemma. We refer readers to this Table on p. 6, right after we introduce the decision structure.

Line 205-7: So how often do people lose all their ties? And how might this impact inference? Apologies if you have described this somewhere else, but I think this is important to share with the reader.

We described network isolates in the Supplementary Information (see esp. pp. 26-27 and pp. 46-48). There we discuss how this happened (some participants dropped out for various reasons [e.g., internet connectivity], others were excluded). We see now that we failed to give readers a heads up in the Main Text that they could find these results in the SI. We now do so on p. 13. (We would be happy to summarize these findings in the main text. The reason we only include this information in the SI is that we largely replicate past work.)

Closely related to the previous point, why are there so few Republicans in Figure 4b “parochial network” (lower right-hand figure)? Is that b/c they lost all their ties? This seems worth addressing.

This is an important question, and deserves clarification. We have added an explanation in the Figure 4 legend. It reads “The networks in B were selected for having segregation patterns typical of the estimated means from A. They are not representative of other patterns within conditions. For instance, while the Inter/Intragroup network in B has fewer Republicans than the Control and Colorblind networks, this is not true of this condition more generally. Indeed, Figure S25 shows that Republicans are less likely to become isolated from the networks in the Inter/Intragroup condition than the two other conditions. See Figures S28-S31 for visualizations of all networks in our experiment, broken down by condition.”

Reviewers' Comments:

Reviewer #2:

Remarks to the Author:

The most recent revision handles my concerns appropriately. The text has improved significantly and ambiguities have been corrected. On my behalf, there is green light for publication. Congratulations for the authors for the hard work.

Reviewer #4:

Remarks to the Author:

I want to thank the authors again for engaging so diligently with my comments in this most recent round of revisions. In particular, I appreciate that they took my concern about the real-world application/implications of the study on social media platforms – including the idea of civility scores – so seriously. I also want to thank the authors for including Table 2, which immediately clarified why their game was a form of the Prisoner's Dilemma, something that was not clear to me from reading earlier versions of the manuscript. I think this change will make the manuscript more accessible to more readers.

I recommend that the paper be accepted for publication at this point, and there is no need on my part for another round of reviews. I have two very minor suggestions that the authors might want to incorporate in their final revisions:

p.8, paragraph 3: the authors note that partisanship played no role in whether people were more likely to accept tie proposals. Given everything else going on in their study, this seems kind of interesting and perhaps worth a sentence speculating why this is the case? It seems to me that it points to the importance of the fact that the game has monetary incentives and that once there is a clear possibility of losing money b/c of partisan bias when there is relevant information (e.g., how the proposer has behaved previously) that speaks to the possibility of that accepting/rejecting the proposal from a financial perspective, the issue of partisan bias is jettisoned. Is that a fair take? And, if so, why is it the case that people rely on partisan bias at other points in the study? Is that because they simply have less information available relative to the situation in which a tie is proposed?

p.13, paragraph 3 (Methods): Might be worth a short explanation as to why the authors used a 1-6 scale for partisanship when the dominant scale for self-reported partisanship in the political science literature is a 1-7 scale from branching questions that includes "independent" as 4. I assume this was because the authors would have had to drop self-identified independents from the study, but this is going to jump out at political scientists who read the Methods section.

Revision Memo

“Reputations for treatment of outgroup members can prevent the emergence of political segregation in cooperative networks”

We were very happy to hear that our paper has been conditionally accepted at *Nature Communications*. The Editor’s suggestions and comments were extremely helpful. As explained in our cover letter, how we responded to the Editor’s many comments and suggestions is best seen with tracked changes versions of the document. At any point in the document where a response from us to an editorial comment or question was in order or helpful, we simply provided it directly in response to the Editor’s specific comments. Thus, this revision memo focuses on the reviewers’ final comments and suggestions.

Below, we reproduce the Reviewer comments verbatim and respond to them in indented, *italicized*, text. For simplicity, we also include comments in the marked-up documents indicating where we respond to the reviewer’s comments.

REVIEWERS' COMMENTS

Reviewer #2 (Remarks to the Author):

The most recent revision handles my concerns appropriately. The text has improved significantly and ambiguities have been corrected. On my behalf, there is green light for publication. Congratulations for the authors for the hard work.

We really appreciate the guidance Reviewer 2 gave us in our previous revisions.

Reviewer #4 (Remarks to the Author):

I want to thank the authors again for engaging so diligently with my comments in this most recent round of revisions. In particular, I appreciate that they took my concern about the real-world application/implications of the study on social media platforms – including the idea of civility scores – so seriously. I also want to thank the authors for including Table 2, which immediately clarified why their game was a form of the Prisoner’s Dilemma, something that was not clear to me from reading earlier versions of the manuscript. I think this change will make the manuscript more accessible to more readers.

I recommend that the paper be accepted for publication at this point, and there is no need on my part for another round of reviews. I have two very minor suggestions that the authors might want to incorporate in their final revisions:

We appreciate the positive comments, and likewise appreciate the helpful suggestions Reviewer 4 offered on previous versions of our paper.

p.8, paragraph 3: the authors note that partisanship played no role in whether people were more likely to accept tie proposals. Given everything else going on in their study, this seems kind of interesting and perhaps worth a sentence speculating why this is the case? It seems to me that it points to the importance of the fact that the game has monetary incentives and that once there is a clear possibility of losing money b/c of partisan bias when there is relevant information (e.g., how the proposer has behaved previously) that speaks to the possibility of that accepting/rejecting the proposal from a financial perspective, the issue of partisan bias is jettisoned. Is that a fair take? And, if so, why is it the case that people rely on partisan bias at other points in the study? Is that because they simply have less information available relative to the situation in which a tie is proposed?

We agree that it is important to call attention to the fact we do not find ingroup preferences with respect to accepting tie proposals, whereas we do find ingroup preferences for other decisions throughout the study. We think the likely explanation for this is one that we'd already introduced for a different reason on p. 11 of the paper with tracked changes. Specifically, we suggest that a tie request from an outgroup member provides individuating information about that group member, thereby reducing the tendency for the person receiving the tie to act according to group identities (i.e., participants receiving tie requests no longer make distinctions between tie requests from ingroup versus outgroup members). This suggestion is in line with prior theory and research we cite in the preceding paragraph of the paper and can be tested in future work.

p.13, paragraph 3 (Methods): Might be worth a short explanation as to why the authors used a 1-6 scale for partisanship when the dominant scale for self-reported partisanship in the political science literature is a 1-7 scale from branching questions that includes "independent" as 4. I assume this was because the authors would have had to drop self-identified independents from the study, but this is going to jump out at political scientists who read the Methods section.

We appreciate this suggestion. The reviewer is correct that we did this to avoid having to drop self-identified independents from the study. We now flag that we used a six-point scale in the Methods section (p. 12 of paper with tracked changes), explain why we did this, and outline the potential implications of using a six-point scale versus a seven-point scale.